# The acclimative biogeochemical model of the southern North Sea

Onur Kerimoglu[1], Richard Hofmeister[1], Joeran Maerz[1,2], Rolf Riethmüller[1], and Kai W. Wirtz[1]

[1]Institute of Coastal Research, Helmholtz-Zentrum Geesthacht, Geesthacht, Germany
[2](present) Max Planck Institute for Meteorology, Hamburg, Germany

*Correspondence to:* Onur Kerimoglu (kerimoglu.o@gmail.com)

**Abstract.** Ecosystem models often rely on heuristic descriptions of autotrophic growth that fail to reproduce various stationary and dynamic states of phytoplankton cellular composition observed in laboratory experiments. Here, we present the integration of an advanced phytoplankton growth model within a coupled 3-dimensional physical-biogeochemical model, and the application of the model system to the Southern North Sea (SNS) defined on a relatively high resolution ($\sim$ 1.5-4.5 km) curvilinear grid. The autotrophic growth model, recently introduced by Wirtz and Kerimoglu (2016), is based on a set of novel concepts for the allocation of internal resources and operation of cellular metabolism. The coupled model system consists of the general estuarine transport model (GETM) as the hydrodynamical driver, a lower trophic level model and a simple sediment diagenesis model. We force the model system with realistic atmospheric and riverine fluxes, background turbidity caused by suspended particulate matter and open ocean boundary conditions. For a simulation for the period 2000-2010, we show that the model system satisfactorily reproduces the physical and biogeochemical states of the system within the German Bight characterized by steep salinity, nutrient and chlorophyll gradients, as inferred from comparisons against observation data from long-term monitoring stations, sparse in-situ measurements, continuous transects, and satellites. The model displays skill also in capturing the formation of thin chlorophyll layers at the pycnocline, frequently observed within the stratified regions during summer. A sensitivity analysis reveals that the vertical distributions of phytoplankton concentrations estimated by the model can be qualitatively sensitive to the description of the light climate and dependence of sinking rates on the internal nutrient reserves. A non-acclimative (fixed-physiology) version of the model predicted entirely different vertical profiles, suggesting that accounting for physiological flexibility might be relevant for a consistent representation of the vertical distribution of phytoplankton biomass. Our results point to significant variability in cellular chlorophyll to carbon ratio (Chl:C) across seasons and the coastal to offshore transition. Up to 3 fold higher Chl:C at the coastal areas in comparison to those at the offshore areas contribute to the steepness of the chlorophyll gradient. The model predicts also much higher phytoplankton concentrations at the coastal areas in comparison to its non-acclimative equivalent. Hence, findings of this study provide evidence for the relevance of the physiological flexibility, here reflected by spatial and seasonal variations in Chl:C, for a realistic description of biogeochemical fluxes, particularly in the environments displaying strong resource gradients.

Wirtz, K.W. and Kerimoglu, O.: Optimality and variable co-limitation controls autotrophic stoichiometry, Frontiers in Ecology and Evolution, doi:10.3389/fevo.2016.00131, 2016.

# 1  Introduction

Modelling the biogeochemistry of coastal and shelf systems requires the representation of a multitude of interacting processes, not only within the water but also at the adjacent earth system components such as the atmosphere (e.g., nitrogen deposition), land (e.g., riverine inputs), sediment (e.g., diagenetic processes), and biochemical processes in water (see., e.g., Cloern et al.,
2014; Emeis et al., 2015). For being able to reproduce the large scale spatial and temporal distribution of biogeochemical variables in coastal systems, a realistic representation of hydrodynamical processes is often critically important, at least those relevant to the circulation patterns and stratification dynamics: the former is needed to describe the spread of nutrient-rich river plumes and exchange at the open ocean boundaries, and the latter for being able to capture the vertical gradients in the light and nutrient conditions for primary productivity. Representation of biological processes and the two way interactions between
biological, chemical and benthic compartments in models are particularly challenging, given the complexity of physiological processes displayed by individual organisms, e.g., regarding the regulation of their internal stoichiometries (e.g., see Bonachela et al., 2016) and the differences in functional traits of species constituting communities (e.g., see Litchman et al., 2010).

3-D ecosytem models often describe the processes relevant to primary production, e.g., the nutrient and light limitation of phytoplankton, using heuristic formulations that have been shown to be inadequate in reproducing patterns obtained in
laboratory experiments. For instance, light limitation is determined not only by the instantaneously available irradiance, but also by the amount of light harvesting apparatus, i.e., chlorophyll pigments maintained by the phytoplankton cells, which can change considerably through a process referred to as photoacclimation. However, photoacclimation is often completely ignored in 3-D model applications, or its effects are mimicked heuristically, for instance, by describing the chlorophyll to carbon ratio as a function of irradiance (Blackford et al., 2004; Fennel et al., 2006), which cannot capture the dependence
of chlorophyll synthesis on nutrient availability (e.g., Pahlow and Oschlies, 2009; Smith et al., 2011; Wirtz and Kerimoglu, 2016). Similarly, interaction of limitation by different nutrient elements is described by heuristic formulations, dichotomously either by a product rule or a threshold function, which, again, cannot reproduce complex patterns observed in laboratory conditions, such as the asymmetric cellular N:C and P:C ratios emerging under N- and P- limited conditions (Bonachela et al., 2016; Wirtz and Kerimoglu, 2016). Such simplifications in the description of primary production processes, in turn, potentially
lead to flawed representations of nutrient cycling. Despite the recently revived theoretical work on stoichiometric regulation and photoacclimation (e.g., Klausmeier et al., 2004; Pahlow and Oschlies, 2009; Wirtz and Pahlow, 2010; Bonachela et al., 2013; Daines et al., 2014), an implementation of a model with a mechanistic description of the regulation of phytoplankton composition at a full ecosystem scale in a coupled physical-biological modeling framework remains to be lacking. In this study, we therefore present a 3-D application of the Model for Adaptive Ecosytems for Coastal Seas (hereafter MAECS), to
the Southern North Sea (SNS), for a decadal hindcast simulation. MAECS features an photoacclimative autotrophic growth model that has been recently introduced by Wirtz and Kerimoglu (2016), which resolves the regulation of the stoichiometry and composition of autotrophs employing an innovative suit of adaptive and optimality based approaches.

The SNS is part of a shallow shelf system (Fig. 1). Especially the south eastern portion of the SNS, known as the German Bight surrounded by the inter-tidal Wadden Sea, is characterized by steep gradients with respect to both nutrients (Hydes et al.,

1999; Ebenhöh, 2004) and turbidity. The latter is largely determined by suspended particulate matter (SPM) concentrations (Tian et al., 2009; Su et al., 2015). These gradients are driven by a complex interplay of riverine and atmospheric fluxes, complex topography, residual tidal currents, density gradients, biological processing of organic matter, benthic-pelagic coupling and sedimentation/resuspension dynamics (Postma, 1961; Puls et al., 1997; van Beusekom and de Jonge, 2002; Burchard et al.,
2008; Hofmeister et al., 2016; Maerz et al., 2016). A number of modelling studies previously addressed the biogeochemistry of the North Sea, including the German Bight. In a majority of these studies, such as ECOHAM-HAMSOM (Pätsch and Kühn, 2008), NORWECOM (Skogen and Mathisen, 2009), ECOSMO-HAMSOM (Daewel and Schrum, 2013), HAMOCC-MPIOM (Gröger et al., 2013), ERSEM-NEMO (Edwards et al., 2012; Ford et al., 2017), ERSEM-POLCOMS (de Mora et al., 2013; Ciavatta et al., 2016) and ERSEM-BFM-GETM (van Leeuwen et al., 2015; van der Molen et al., 2016), large domains and
relatively coarse grids were employed ($\geq 7$ km). While showing good skill in reproducing offshore dynamics, these models seemed to have a relatively limited performance at the shallow, near-coast regions (when reported). The BLOOM-Delft3D (Los et al., 2008) on the other hand, is one of the rare examples with a finer grid (down to 1 km at the Dutch coasts) at the cost of a relatively smaller domain, similar to ours. Although this model system performs decently at both coastal and offshore areas, its performance within the German Bight has not been fully assessed. Moreover, none of these models provide elaborate
descriptions of the stoichiometric regulation of autotrophs, as mentioned above. Therefore, our new model system is expected to fill two important gaps by;

1. Exemplifying for the first time to the best of our knowledge, implementation of a highly complex phytoplankton growth model at an ecosystem scale, coupled to a hydrodynamic model and other biogeochemical compartments, and gain some first insight into the relevance of acclimation to the modelling of coastal biogeochemistry.

2. Establishing the capacity to reproduce the biogeochemistry of the German Bight both at coastal and offshore regions with a single parameterization and model setup.

For a 11 year hindcast simulation of the period 2000-2010, we show that the model can adequately capture the spatio-temporal variability of the physical and biogeochemical features of the SNS based on comparisons against various data sources. Importantly, the model can reproduce the steep chlorophyll and nutrient gradients prevalently observed across the Wadden Sea-
German Bight continuum. We show that the chlorophyll gradients are linked with nutrient, hence, productivity gradients, and further amplified by acclimation capacity of the phytoplankton, and particularly by the high chlorophyll to carbon ratios at the coastal regions.

## 2  Methods

### 2.1  Observations

Observation data from Helgoland Roads, Sylt and 17 other monitoring stations reflect surface measurments. Extensive analyses of the data from Helgoland Roads have been previously performed by Wiltshire et al. (2008) and from Sylt by Loebl et al.

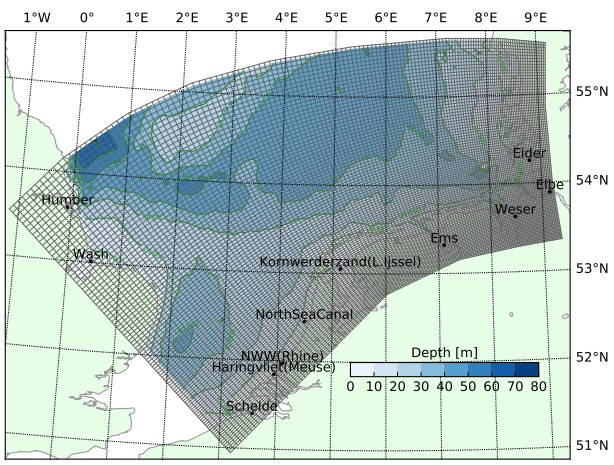

**Figure 1.** Bathymetry of the model domain and the location of rivers considered in this study. Gray lines display the model grid.

(2007). Sparse measurements of temperature, salinity, dissolved inorganic nitrogen, phosphorus and chlorophyll were obtained from the online database of International Council for the Exploration of the Sea (ICES, www.ices.dk).

Continuous Scanfish and FerryBox measurements were performed within the operation of the Coastal Observing System for Northern and Arctic Seas (COSYNA, Baschek et al., 2016). Data collection, processing and quality control of the Scanfish data are described by Maerz et al. (2016) and of the FerryBox data by Petersen (2014). Satellite dataset used here is the Ocean Colour Climate Change Initiative, Version 3.1, European Space Agency, available online at http://www.esa-oceancolour-cci.org/. Chlorophyll estimates of the satellite product were bias-corrected according to the product user guide (Grant et al., 2017): $C_{bc} = 10^{\log_{10}(C) + \delta}$, where $C_{bc}$, $C$ and $\delta$ are, respectively, the bias-corected, raw, and log10 bias estimates for chlorophyll concentrations.

## 2.2 Model

Major processes taken into account by the model are the lower trophic food web dynamics, phytoplankton ecophysiology and basic biogeochemical transformations in the water, and the transformation of N- and P-species in the benthos (Fig. 2 and Section 2.2.1). Physical processes are resolved by the coupled 3-D hydrodynamical model, GETM (Section 2.2.2). Turbidity caused by suspended particulate matter (SPM), nutrient loading by rivers and atmospheric nitrogen deposition were considered as model forcing (Section 2.2.3). The model grid and rivers considered in this study are shown in Fig. 1.

### 2.2.1 Biogeochemical model

The pelagic module, the Model for Adaptive Ecosystems in Coastal Seas (MAECS), is a lower trophic level model that resolves cycling of carbon, nitrogen (N) and phosphorus (P), and importantly, acclimation processes involved in phytoplankton growth. In MAECS, the acclimation of phytoplankton is resolved by a scheme recently introduced by Wirtz and Kerimoglu (2016),

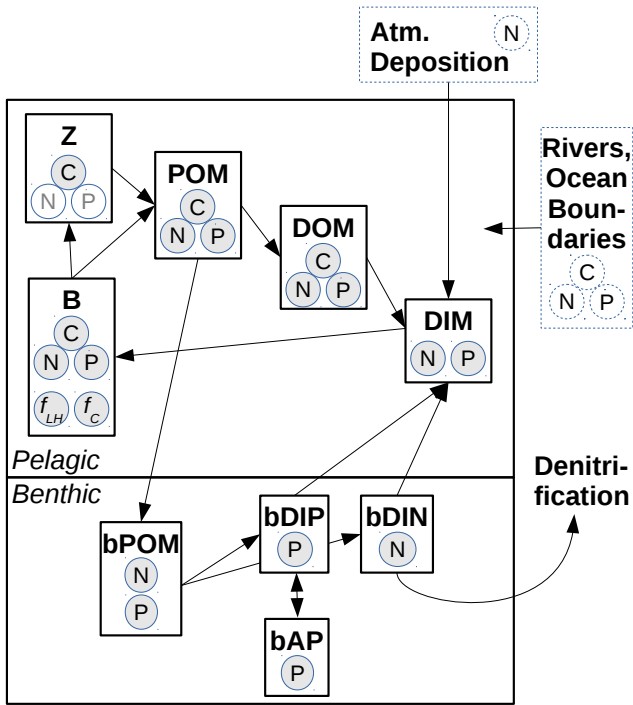

**Figure 2.** Structure of the biogeochemical model. Model components (rectangles) comprise; B: phytoplankton, Z: zooplankton, POM and DOM: particulate and dissolved organic matter, DIM(N,P): dissolved inorganic matter (nitrogen,phosphorus), b-AP: P adsorbed in iron-phosphorus complexes (See Section 2.2.1 and Appendix A for further details). C, N, P in small circles refer to carbon, nitrogen and phosphorus bound to each component, respectively, whereas $f_{LH}$ and $f_C$ are the allocation coefficients for light harvesting and carboxylation (Section 2.2.1). Boxes in dashed lines indicate model forcing.

which describes the instantaneous or transient optimization of physiological traits, $x$ by the extended optimality principle:

$$\frac{\mathrm{d}}{\mathrm{d}t}x = \delta_x \cdot \left[ \frac{\partial V_C}{\partial x} + \sum_i \frac{\partial V_C}{\partial q_i}\frac{\partial q_i}{\partial x} \right] \tag{1}$$

where $\delta_x$ corresponds to the flexibility of traits (Eq.A18), $i$ expands to N and P, and the two terms in brackets describe the direct effects of trait changes on the specific phytoplankton growth rate $V_C$ (in units of cellular C) and the indirect effects through changes in the Chl:C:N:P stoichiometry, expressed by the quotas $q$, respectively. Specifically, three-levels of acclimative regulations are considered (see Fig. 2 in Wirtz and Kerimoglu, 2016):

1. Machinery allocation: we describe the changes in allocations to light harvesting, carbon fixation and nutrient acquisition machineries, as also in (Wirtz and Pahlow, 2010). These allocations correspond to synthesis of cellular structures like chloroplasts for absorbing light, Rubisco enzyme involved in carboxylation process and proteins for gathering nutrient molecules, therefore we track these fractional allocations with two dynamic state variables, $f_{LH}$ and $f_C$, that describe the allocations for light harvesting and carboxylation, while the allocations for nutrient uptake, $f_V$ is assumed to be the

rest $1 - f_{LH} - f_C$. Here, the flexibility term, set to $\delta_x = f_x \cdot (1 - f_x)$, regulates the speed of optimization as determined by the differential terms in Eq.1.

2. Nutrient affinity-processing optimality: we assume that there is a trade-off between nutrient affinity and processing, and the optimal affinity fractions for each nutrient, $f_i^A$, are instantaneously optimized, such that $d_x/d_t = 0$ and $f_i^A$ are algebraically found by setting $\frac{\partial V_C}{\partial x} = 0$ (Pahlow, 2005; Smith et al., 2009).

3. Nutrient uptake activity: (down-) regulation of the uptake rate of nutrients, which is often formulated as a linear function of nutrient quotas (Morel, 1987) in traditional models, is in our approach, described by the instantaneously optimized uptake activity trait, $a_i$. Assuming that energy expenditure for taking up of each nutrient depends on the metabolic needs, values of $a_i$ are found by scaling their marginal growth benefits (Eq.A17).

Driven by the variations of these physiological traits, Chl:C:N:P stoichiometry varies continuously depending on ambient light and nutrient conditions and on the metabolic demands of autotrophic cells. As a further novel aspect of the acclimation model, multiple limitation is described as a queuing function, which allows formulating the co-limitation strength as a function of internal nitrogen reserve $q_N$, instead of prescribing it to be either high as by a product rule, or low as by a threshold (Liebig) function (Wirtz and Kerimoglu, 2016). A detailed description of the phytoplankton growth module can be found in Wirtz and Kerimoglu (2016). Equations and parameters of the model are provided in Appendix A1.

Other components of the pelagic module are similar to standard descriptions in state-of-the-art ecosystem models. Phytoplankton take up nutrients in the form of dissolved inorganic material (DIM). Losses of phytoplankton (B) and zooplankton (Z) due to mortality are added to the particulate organic matter (POM) pool, which degrades into dissolved organic material (DOM), before becoming again DIM and closing the cycle (Appendix A1). As a relevant aspect of the model, while the sedimentation speed of POM ($w_{POM}$) prescribed as a constant value, that of the phytoplankton, $w_B$ is assumed to be modified by its nutrient (quota) status. As decreased internal nutrient quotas likely affect the cells ability to regulate buoyancy and lead to faster migration towards deeper, potentially nutrient rich waters (Boyd and Gradmann, 2002), we assume that maximum sinking rates realized at fully depleted quotas converge to a small background value with increasing quotas as has been observed especially for, but not limited to, diatoms (Smayda and Boleyn, 1965; Bienfang and Harrison, 1984). Although the phytoplankton sinking is often parameterized as a constant rate in 3-D modelling applications, similar formulations of increasing sinking rates under nutrient stress have been also used (e.g., Vichi et al., 2007).

The benthic module describes only the dynamics of macronutrients N and P. Degradation of OM to DIM is described as a one step, first order reaction. Denitrification is described as a proportion of POM degradation, limited by DIN and dissolved oxygen (DO) availability in benthos. As DO is not directly modeled, it is estimated from temperature in order to mimic the seasonality of the hypoxia-driven denitrification. The model accounts for the sorption-desorption dynamics of phosphorus as an instantaneous process also as a function of temperature based on the correlation observed in the field (Jensen et al., 1995). Further details are provided in Appendix A2.

### 2.2.2 Hydrodynamic model and model coupling

The General Estuarine Transport Model (GETM) was used to calculate various hydrodynamic processes, as well as the transport of the biogeochemical variables. A detailed description of GETM is provided by Burchard and Bolding (2002) and Stips et al. (2004). GETM utilizes the turbulence library of the General Ocean Turbulence Model (GOTM) to resolve vertical mixing of density and momentum profiles with a k-$\varepsilon$ two equation model (Burchard et al., 2006). GETM was run in baroclinic mode, resolving the 3-D dynamics of temperature, salinity and currents and 2-D dynamics of sea surface elevation and flooding-drying of cells at the Wadden Sea. Following Gräwe et al. (2016), we assumed the bottom roughness length to be constant through out the domain, and $z_0=10^{-3}$m. We used 20 terrain-following layers and a curvilinear grid of 144x98 horizontal cells, providing a horizontal resolution of approximately 1.5 km at the south-east corner and 4.5 km at the north-west corner (Fig. 1). The curvilinear grid focuses on the German Bight, and roughly follows the coastline (Fig. 1) for an optimal representation of along- and across- shore processes. Similar gridding strategies were applied successfully in other coastal setups with the GETM model (Hofmeister et al., 2013; Hetzel et al., 2015). We employed integration time steps of 5 and 360 seconds for the 2-D and 3-D processes, respectively.

Integration of model forcing was realized through the Modular System for Shelves and Coasts (MOSSCO, http://www.mossco.de), which, among others, provides standardized data representations (Lemmen et al., 2017). Meteorological forcing originated from an hourly-resolution hindcast by COSMO-CLM (Geyer, 2014). Boundary conditions for surface elevations are extracted from an hourly resolution hindcast by TRIM-NP (Weisse et al., 2015). For temperature and salinity, daily climatologies from HAMSOM (Meyer et al., 2011) are used, all of which are available through coastDat http://www.coastdat.de.

Two-way coupling of the biological model with GETM was achieved via the Framework for Aquatic Biogeochemical Models (FABM, Bruggeman and Bolding, 2014). The pelagic module is defined in the 3-D grid of the hydrodynamic model, whereas the benthic module is defined in 0-D boxes for each water column across the lateral grid of the model domain (Fig. 1). Each benthic box interacts with the bottom-most pelagic box of the corresponding water column in terms of a uni-directional flux of POM from the pelagic to the benthic states, and a bi-directional flux of DIM depending on the concentration gradients.

For the integration of the source terms, a fourth order explicit Runge-Kutta scheme was used with an integration time step of 360 seconds, as for the 3-D fields in GETM. Exchange between pelagic and benthic variables was integrated with a first order explicit scheme at a time step identical to that of the biological model.

### 2.2.3 Model forcing and boundary conditions

Light extinction is described according to:

$$I(z) = I_0 a e^{-\frac{z}{\eta_1}} + I_0(1-a)e^{-\frac{z}{\eta_2} - \int_z^0 \sum_i k_{c,i} c_i(z')dz'} \tag{2}$$

where, $I_0$ is the photosynthetically available radiation at the water surface, and the first and second terms describe the attenuation at the red and blue-green portions of the spectrum. We assume that the partitioning of the two ($a$) and the attenuation length scale of the red light ($\eta_1$) are constant over space and time as in Burchard et al. (2006), and that the attenuation of

blue-green light is due to SPM (as described by $\eta_2$) and organic matter (sum term). We chose $a = 0.58$ and $\eta_1 = 0.35$, which correspond to Jerlov class-I type water, thus clear water conditions (Paulson and Simpson, 1977), given that the attenuation by SPM and organic matter is explicitly taken into account. For calculating attenuation due to SPM, a daily climatology of SPM concentrations defined over the model domain was utilized, like in ECOHAM (Große et al., 2016). The SPM field was

constructed by multiple linear regression of salinity, tidal current speed and depth for each Julian day (Heath et al., 2002). Then, $\eta_2$, or the inverse of SPM caused attenuation coefficient was calculated according to:

$$1/\eta_2 = k_{SPM} = \epsilon_{SPM} * SPM \tag{3}$$

where, the attenuation for background turbidity, $K_w = 0.16$ m$^{-1}$ and specific attenuation coefficient for SPM, $\epsilon_{SPM} = 0.02$ m$^2$ g$^{-1}$ according to Tian et al. (2009). For calculating the attenuation due to organic matter in Eq.(2), phytoplankton, POC

and DOC were considered (Table A3).

Freshwater and nutrient influxes were resolved for eleven major rivers along the German, Dutch, Belgian and British coasts (Fig. 1). For eight of these rivers, Radach and Pätsch (2007) and Pätsch and Lenhart (2011) presented a detailed quantitative analysis of nutrient fluxes. Besides the fluxes in inorganic form based on direct measurements, fluxes in organic form have been accounted for, first by calculating the total organic material concentration by subtracting dissolved nutrient concentrations

from total nitrogen and total phosphorus, then by assuming 30 % of the organic material to be in particulate form (i.e., POM; Amann et al., 2012). Further, 20 % of POM is assumed to describe phytoplankton biomass (Brockmann, 1994), C:N:P ratio of which was assumed to be in Redfield proportions. Finally, no estuarine retention/enrichment was assumed, following Dähnke et al. (2008). All river data except for the river Eider were available in daily resolution, however with gaps. Short gaps (<28 days) were filled by linear interpolation. Loadings from the river Eider were calculated first by merging the data measured

at the stations on two upstream branches, Eider and Treene, then by filling the short gaps (<28 days) by linear interpolation, replacing the larger gaps with daily climatology, and extending for 2000-2003 by using the climatology as well. To describe DIN deposition at the water surface, sum of annual average atmospheric deposition rates of $NO_x$ and $NH_3$ provided by EMEP (European Monitoring and Evaluation Programme, http://www.emep.int) were used. At the open boundaries in the north and west of the model domain (Figure 1), all state variables belonging to the phytoplankton and zooplankton compartments are

assumed to be at zero-gradient. For DIM, DOM and POM, monthly values of ECOHAM (Große et al., 2016), interpolated to 5m depth intervals are used as clamped boundary conditions.

## 2.3 Quantification of Model Performance

For the comparisons with the data at monitoring stations, sparse in-situ measurements from the ICES database, and with the satellite (ESA-CCI oceancolor) dataset, Pearson correlation coefficients, $\rho$, and mean normalized bias, $B^* = (\langle S \rangle - \langle O \rangle)/\langle O \rangle$,

where $\langle S \rangle$ and $\langle O \rangle$, respectively, are the average simulated and observed values) were calculated. For the DIN, DIP and chlorophyll comparisons with the station and ICES data, these skill scores are reported in a color-coded table, where the 4 color levels indicate low (red: $|B*| \geq 0.75$ and $\rho < 0.25$), moderate-low (yellow: $0.5 \geq |B*| < 0.75$ and $0.25 \leq \rho < 0.5$), moderate-high (green: $0.25 \geq |B*| < 0.5$ and $0.5 \leq \rho < 0.75$) and high (blue $|B*| \leq 0.25$ and $\rho \geq 0.75$) model performance. For the comparisons

against the sparse ICES and ESA-CCI data, correlation scores and model standard deviations normalized to measured standard deviations are displayed as Taylor diagrams, where the correlation score and the normalized standard deviation correspond to the angle and distance to the center (Jolliff et al., 2009). For the comparisons against the ICES and ESA-CCI data, only the middle-99 percentile of simulated and measured values were considered (i.e., leaving out the first and last 0.5th percentiles).

For the ICES data, temporal matching was identified at daily resolution, vertical matching was obtained by comparing the measurements within the upper 5 meters from the sea surface and within the 5 meters above the sea floor with the model estimates at the top-most and bottom-most layers, and finally horizontal matching by calculating the average of the values from four nearest cells surrounding the measurement location, inversely weighted by their Cartesian distance. For the ESA-CCI data, the temporal matching was obtained by averaging the data from both sources for the period 2008-2010 for particular seasons

of the year, and horizontal matching by performing a 2-dimensional linear interpolation of the satellite data to the model grid. Extraction of the hourly model temporally matching to the Scanfish data was based on the hourly-binned average time for each cast (defined as a full downward and upward undulation cycle) and 3-D spatial matching was obtained by constructing an average vertical profile from the 4 closest cells to the average coordinate of each cast. For facilitating the qualitative comparison of the simulated chlorophyll and the Scanfish measurements of fluorescence, which have different units and signal strengths,

normalized anomalies were used, according to $\hat{p}_i = (p_i - \langle p \rangle)/\sigma_p$, where $\hat{p}_i$ and $p_i$ are the normalized anomaly and raw value of a given data point, and $\langle p \rangle$ and $\sigma_p$ are the mean and standard deviation of all data points.

## 3   Results

### 3.1   Evaluation of Model Performance by in-situ Data

A comparison of simulated salinities with the FerryBox measurements along the cruise between Cuxhaven (at the mouth

of river Elbe) and Immingham (at the mouth of river Humber), demonstrates that the model captures the horizontal salinity distribution (Fig. 3). In particular, the contrast between the north-western model domain characterized by the rapid flushing of the coastal freshwater input and the south-eastern model domain (i.e., German Bight) characterized by a strong and permanent salinity gradient is well captured. Confinement of the salinity front during winter towards the coast and its seaward intrusion especially during early spring, and the smaller scale modulations that appear to be controlled by the spring-neap cycle are both

reproduced by the model.

    Comparison of simulated surface and bottom temperatures with those extracted from the ICES dataset for the period 2006-2010 are provided in Fig. 4. High correlation scores and low bias attained for water temperature and salinity suggest that the model can reproduce the seasonal warming, spread of freshwater discharges and thermohaline stratification dynamics. However, in a relatively small number of instances, surface temperatures are underestimated and bottom temperatures are

overestimated, which indicates that not all stratification events were captured. Almost all of these instances are found to be located either at the north-eastern margin ($> 4°$E $\&$ $> 55°$N) and at the north-western corner ($< 4°$E $\&$ $> 54°$N) of the model domain, i.e., close to the open ocean boundary (Fig. 1).

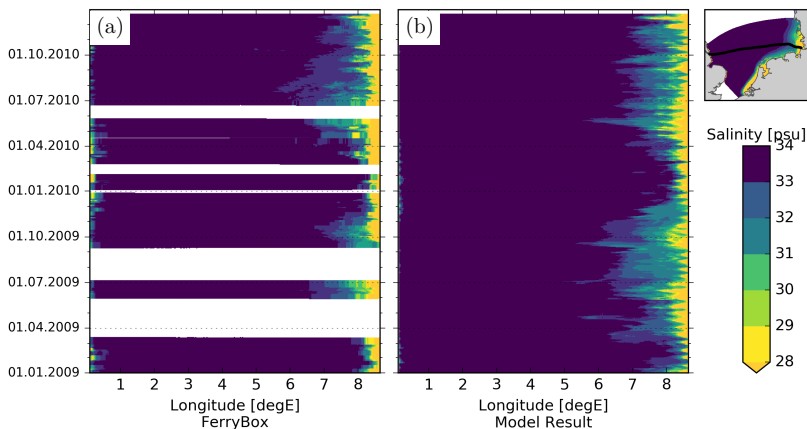

**Figure 3.** Salinity [PSU] measured by FerryBox (a) and estimated by the model (b) along the route shown in the inset. Note that the lower range of salinity was truncated

Comparisons of surface chlorophyll, DIN and DIP concentrations estimated by the model with the measurements in 19 stations scattered across the southern North Sea are shown in Fig. 5-7, and the corresponding skill scores are listed in Table 1. Estimates of average nutrient concentrations, and the timing of their depletion and regeneration in a majority of stations agree well with the observations, as indicated by the frequency of 'high' and 'moderate-high' scores (Table 1). Notably, at several

stations (e.g., Sylt,T8,T36,T26,T22,T11,T12) the difference between the relative bias for DIP and DIN (i.e., $B_{DIP}^* - B_{DIN}^*$) was relatively large (with 55% being highest at T22), suggesting a tendency for underestimating the DIN:DIP ratio, although this was not the case for the comparison against ICES measurements (see below). Relative to the nutrients, performance of the model in estimating the chlorophyll is lower, especially at the stations located along the Dutch coast (Fig. 6, Table 1). However, for about half of the 10 stations where data is available, the model performance is at moderate levels.

Comparison of model results with the DIN, DIP and chlorophyll measurements available at the ICES database at the surface and bottom layers for the entire simulation period indicate negligible normalized mean bias and correlation coefficients at around 0.6%–0.7% for nutrients and about 60% overestimation and correlation coefficients of about 0.3 for chlorophyll (Fig. 8, Table 1). Modeled variability for all three bigeochemical state variables is within an approximately 50% envelope of the observed variability (Fig. 8).

For an assessment of the accuracy of the simulated vertical distributions, water density (expressed as $\sigma_T$) and fluorescence captured by a Scanfish cruise (Heincke-331) obtained during 13-19 July 2010 were compared to those estimated by the model (chlorophyll for fluorescence) averaged over the same time period (Fig. 9). This period was characterized by significant thermal summer stratification reaching deep into the near coastal regions of the German Bight. Thus, $\sigma_T$ reflects two major mechanisms that control the distribution of phytoplankton: first is the vertical gradients characterized by denser water at the bottom layers,

which is mainly driven by thermal stratification as suggested by temperature profiles (not shown). Second is the horizontal gradients characterized by lighter water at the coasts, driven by low salinity due to the freshwater flux from the rivers. The

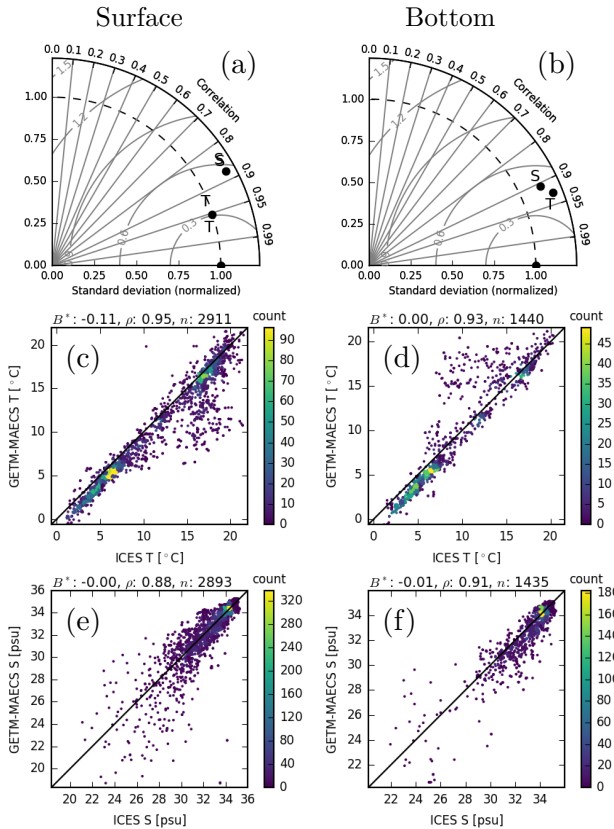

**Figure 4.** Comparison of modeled and measured (ICES) temperature (abbreviated T in panels a,b,c,d) and salinity (S in a,b,e,f) at the surface (left) and bottom (right) layers for the period 2006-2010. 2-D histograms show the number of occurrence of simulation-measurement pairs. Normalized bias ($B^*$), Pearson correlation coefficients ($\rho$), and corresponding number of data points ($n$) are shown on top of scatter plots.

model can accurately reproduce both vertical and horizontal density gradients, although some discrepancies exist, such as slightly underestimated depth of the pycnocline and steepness of lateral gradients at around the coastal section. Fluorescence measurements along the Scanfish track in July 2010 indicate frequent occurrences of subsurface chlorophyll maxima (Fig. 9). These are in some cases in the form of higher concentrations below the pycnocline but in some others, appear as thin layers at around the pycnocline. While the deep chlorophyll maxima are prevalently found in stratified offshore regions, the well-mixed shallower regions mostly show homogeneously distributed high chlorophyll concentrations throughout the water column due to higher dissipation rates (Maerz et al., 2016). The MAECS simulation agrees qualitatively well with these patterns and captures the spatial variability of the observed vertical chlorophyll distribution (Fig. 9).

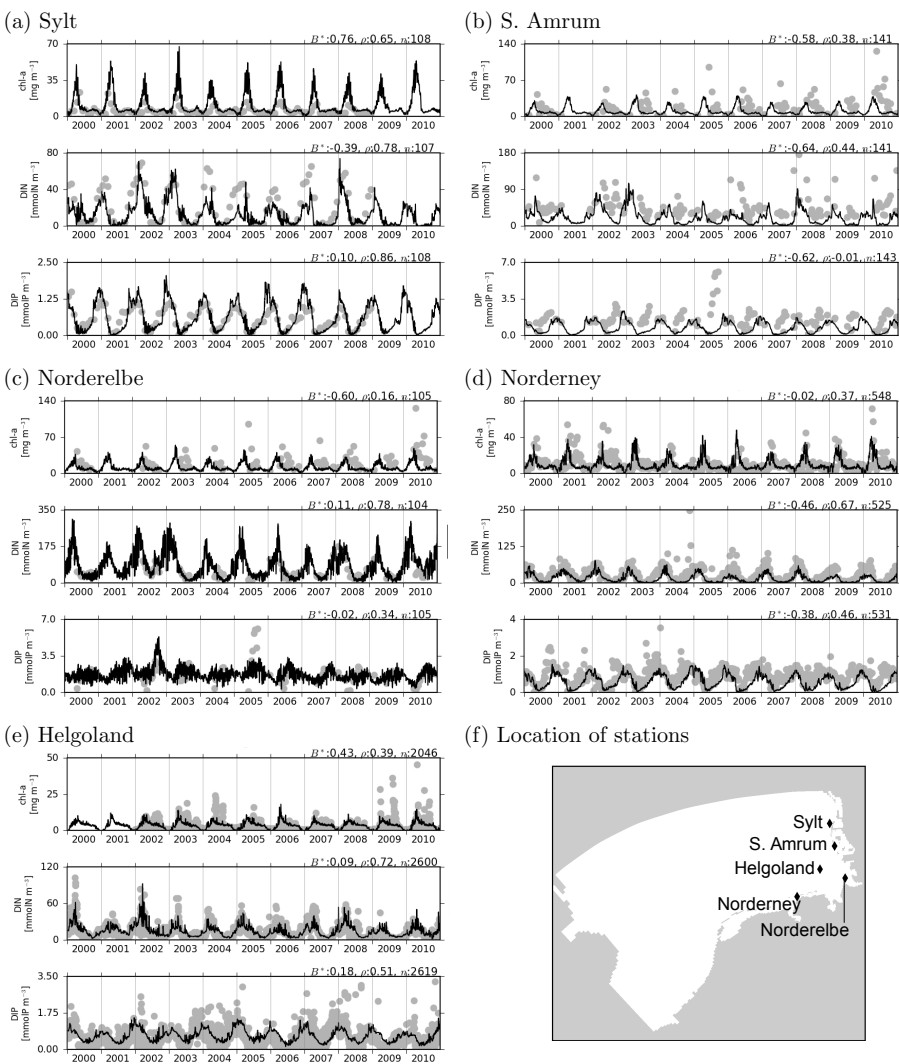

**Figure 5.** Observations (gray dots) and model estimates (lines) of surface chlorophyll, DIN and DIP concentrations at the stations located along the coasts of the German Bight, operated by Alfred Wegener Institute (Helgoland and Sylt), Landesamt für Landwirtschaft, Umwelt und ländliche Räume des Landes Schleswig-Holstein (S. Amrum, Norderelbe) and Niedersächsischer Landesbetrieb für Wasserwirtschaft, Küsten- und Naturschutz (Norderney). Normalized bias ($B^*$), Pearson correlation coefficients ($\rho$), and corresponding number of data points ($n$) are shown on top of each panel.

## 3.2   Coastal Gradients

Temperature stratification is one of the key drivers of biogeochemical processes through its determining role on the resource environment, i.e., light and nutrient availability experienced by the primary producers. The comparison against Scanfish transect (Fig. 9) showed that the physical model has the potential to realistically capture the density stratification. Using the tempera-

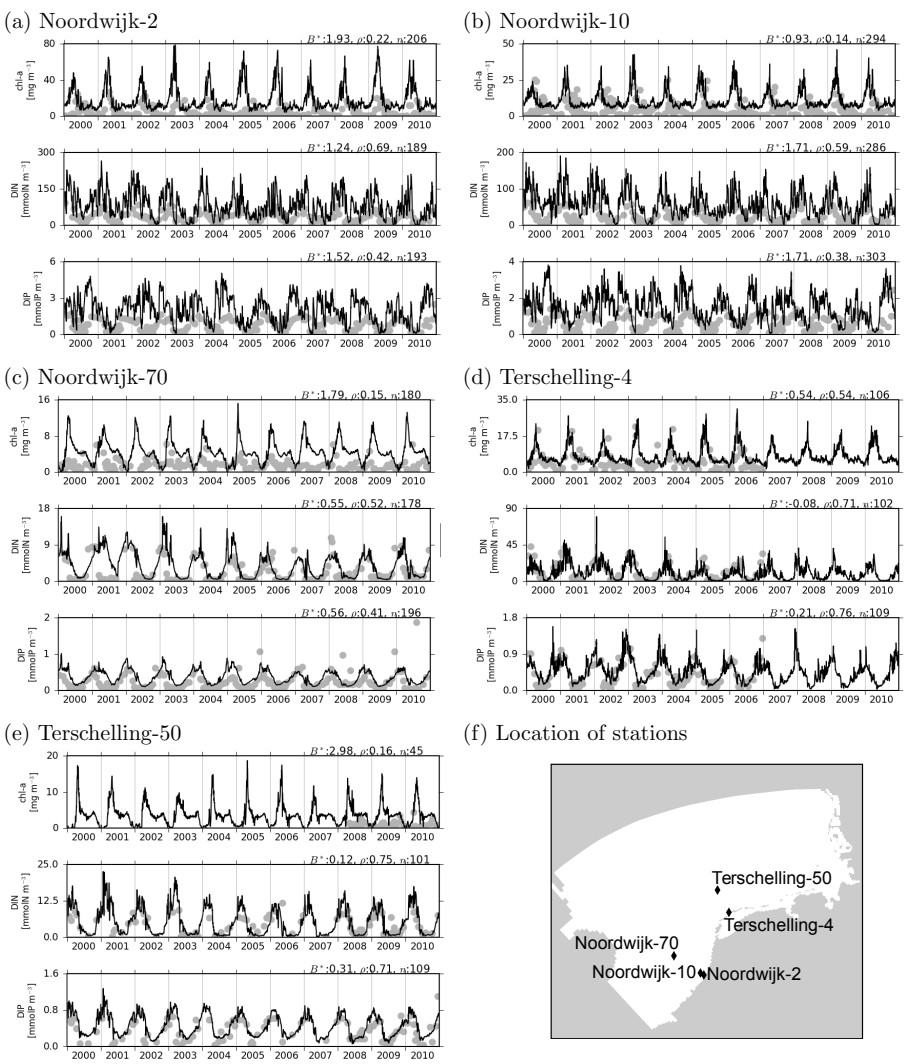

**Figure 6.** As in Fig. 5, but for the stations located along the coasts of the Netherlands, operated by Rijkswaterstaat.

ture difference between surface and bottom layers as an indicator of temperature stratification (Schrum et al., 2003; Holt and Umlauf, 2008; van Leeuwen et al., 2015), and using monthly averages across all simulated years (2000-2010), the areal extent and seasonality of stratification within the SNS is shown in Fig. 10. This analysis suggests that a large portion of the model domain deeper than ∼30 m becomes stratified from April to September, with a maximum areal coverage and intensity (slightly above 8 K) in July.

Simulated climatological concentrations of DIN and DIP display steep coastal gradients along the coasts of the German Bight (Fig. 11), both during the non-growing season (months 1-3 and 10-12) and the growing season (months 4-9). Within the ROFI (Region of Freshwater Influence, Simpson et al., 1993) of Rhine, nitrogen concentrations decrease about 5 fold (from ≥

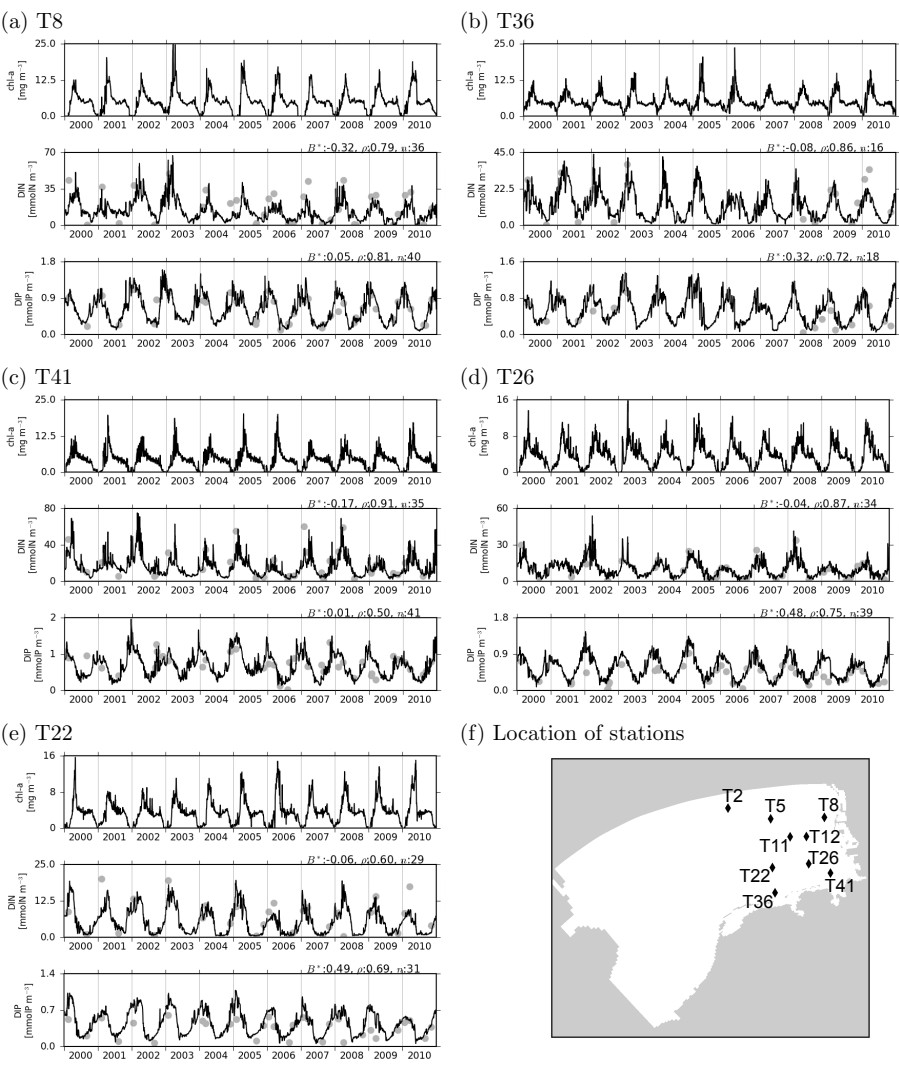

(a) T8    (b) T36    (c) T41    (d) T26    (e) T22    (f) Location of stations

(continued on the next page)

48 mmolN m$^{-3}$ to 8-16 mmolN $^{-3}$) within a few grid cells, corresponding to about 10-15 km distance. In the German Bight, non-growing season is similarly characterized by a thin stripe of high nutrient concentrations along the coast whereas during the growing season, especially phosphorus becomes depleted outside a confined zone of the Elbe plume. At the offshore areas, nutrient concentrations during the growing season are considerably lower than those during the non-growing season, driven by the phytoplankton growth, both directly by nutrient uptake and for the case of nitrogen also indirectly, by fueling denitrification in the sediment. The DIN:DIP ratio in the offshore regions is close to the Redfield molar ratio of 16:1 throughout the year reflecting oceanic conditions, while much higher at the coastal areas, particularly during the non-growing season, reflecting the high N:P content of the continental rivers (Radach and Pätsch, 2007). This transition from high coastal to low offshore N:P ratios is qualitatively consistent with observations (e.g., Burson et al., 2016).

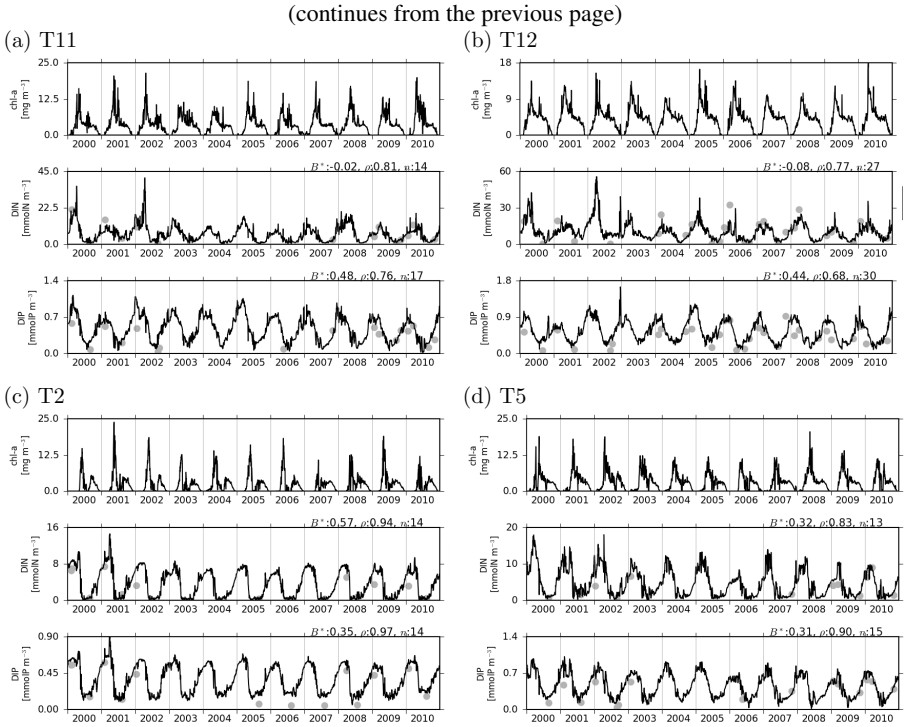

**Figure 7.** As in Fig. 5, but for the offshore monitoring stations operated by Bundesamt für Seeschifffahrt und Hydrographie.

Both the satellite (ESA-CCI) images and our model estimates, averaged again for the non-growing and growing season, suggest steep coastal gradients in chlorophyll concentrations (Fig. 12) similar to the nutrient gradients shown above (Fig.11). The large scale agreement in coastal gradients result in high correlation coefficients (Fig. 12, Table 1). Normalized mean bias is small for the non-growing season but relatively high and positive (i.e., overestimation) for the growing season. Higher model

estimates at the lower range (0-10 mgChl m$^{-3}$) is responsible for this positive bias, which is particularly the case during the first half of the growing season, where the bias is highest (Table 1).

Our simulation results indicate significant spatio-temporal variability in Chl:C ratio, even when the seasonal averages are considered, i.e., omitting short-term variability (Fig. 13). Chl:C ratio is in general higher at the coasts than at offshore. Higher Chl:C ratios during the non-growing (months 10-12 and 1-3) season similarly reflect light limitation due to low amounts of

incoming short wave radiation at the water surface. The simulated spatio-temporal differences in Chl:C ratios reach to about three fold between different seasons of the year and between offshore and coastal areas. The latter suggests that the differential acclimative state of phytoplankton cells amplify the steepness of the chlorophyll gradients across the coastal transition shown in Fig. 12.

For gaining a better understanding of the relevance of acclimation for capturing the coastal gradients, we considered a sim-

plified, non-acclimative version of the model in which the resource utilization traits were fixed (see Appendix B3 for a detailed description), and two alternative parameterizations regarding the allocations to the light harvesting, nutrient acquisition and

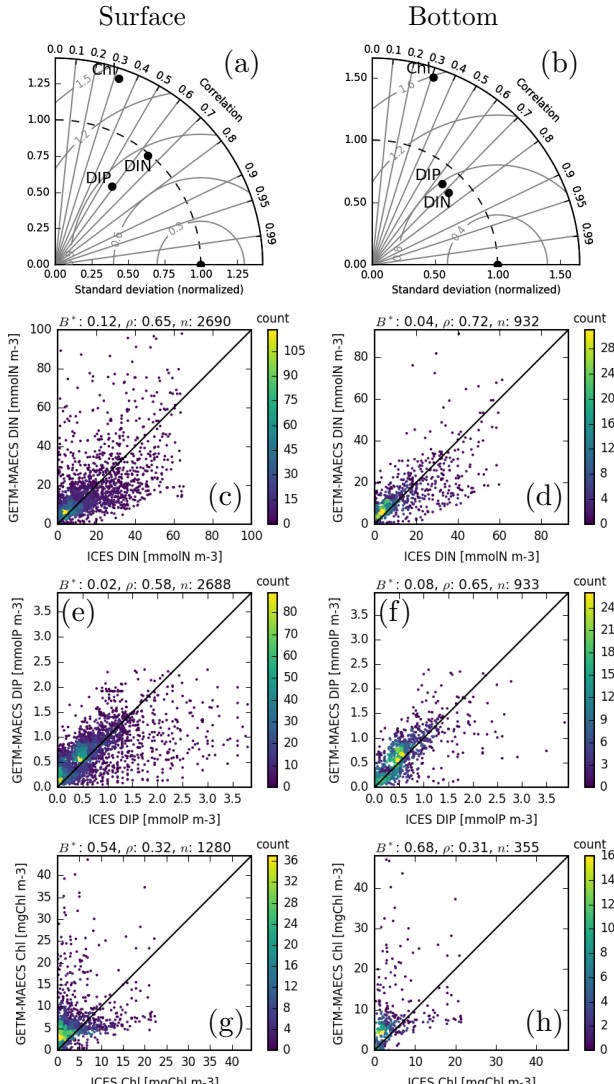

**Figure 8.** Comparison of simulated and measured (ICES) DIN (a,b,c,d), DIP (a,b,e,f) and chlorophyll (a,b,g,h) at the surface (left) and bottom (right) layers for the period 2000-2010. 2-D histograms show the number of occurrence of simulation-measurement pairs. Normalized bias ($B^*$), Pearson correlation coefficients ($\rho$), and corresponding number of data points ($n$) are shown on top of scatter plots.

carboxylation machineries (which are state variables in the full model): first one with equal (balanced) allocation coefficients (=0.333), and a second one by assigning the spatio-temporal averages of the state variables integrated by the full (reference) model. Results of these two parameterizations were almost identical, so hereafter we will refer to the 'fixed' model in short, without specifying the particular parameterization.

5    The annual average coastal phytoplankton concentrations estimated by the acclimative model are much higher than those estimated by the fixed model, with no significant difference between the surface and water column averaged values (Fig.

**Table 1.** Skill scores obtained at each station ($B^*$ normalized bias, $\rho$: Pearson correlation coefficients, and $n$: number of matching data points), against ICES and ESA-CCI data shown in Fig.5-8 and Fig.12 partially (for the averages of months 1-3,10-12 and 4-9). Colors indicate skill level, with red: low, yellow: moderate-low, green: moderate-high, blue: high (see Section 2.3).

| | DIN | | | DIP | | | Chl | | |
|---|---|---|---|---|---|---|---|---|---|
| **Station** | $B^*$ | $\rho$ | $n$ | $B^*$ | $\rho$ | $n$ | $B^*$ | $\rho$ | $n$ |
| Sylt | -0.39 | 0.78 | 107 | 0.10 | 0.86 | 108 | 0.76 | 0.65 | 108 |
| S. Amrum | -0.64 | 0.44 | 141 | -0.62 | -0.01 | 143 | -0.58 | 0.38 | 141 |
| Norderelbe | 0.11 | 0.78 | 104 | -0.02 | 0.34 | 105 | -0.60 | 0.16 | 105 |
| Norderney | -0.46 | 0.67 | 525 | -0.38 | 0.46 | 531 | -0.02 | 0.37 | 548 |
| Helgoland | 0.09 | 0.72 | 2600 | 0.18 | 0.51 | 2619 | 0.43 | 0.39 | 2046 |
| Noordwijk-2km | 1.22 | 0.69 | 189 | 1.50 | 0.42 | 193 | 1.92 | 0.22 | 206 |
| Noordwijk-10km | 1.70 | 0.59 | 286 | 1.71 | 0.38 | 303 | 0.94 | 0.14 | 294 |
| Noordwijk-80km | 0.49 | 0.54 | 178 | 0.53 | 0.42 | 196 | 1.82 | 0.17 | 180 |
| Terschelling-4km | -0.10 | 0.70 | 102 | 0.19 | 0.77 | 109 | 0.47 | 0.54 | 106 |
| Terschelling-50km | 0.12 | 0.76 | 101 | 0.31 | 0.72 | 109 | 3.01 | 0.17 | 45 |
| T36 | -0.08 | 0.86 | 16 | 0.32 | 0.72 | 18 | - | - | 0 |
| T26 | -0.04 | 0.87 | 34 | 0.48 | 0.75 | 39 | - | - | 0 |
| T41 | -0.17 | 0.91 | 35 | 0.01 | 0.50 | 41 | - | - | 0 |
| T8 | -0.32 | 0.79 | 36 | 0.05 | 0.81 | 40 | - | - | 0 |
| T2 | 0.57 | 0.94 | 14 | 0.35 | 0.97 | 14 | - | - | 0 |
| T22 | -0.06 | 0.60 | 29 | 0.49 | 0.69 | 31 | - | - | 0 |
| T5 | 0.32 | 0.83 | 13 | 0.31 | 0.90 | 15 | - | - | 0 |
| T12 | -0.08 | 0.77 | 27 | 0.44 | 0.68 | 30 | - | - | 0 |
| T11 | -0.02 | 0.81 | 14 | 0.48 | 0.76 | 17 | - | - | 0 |
| ICES–surface | 0.12 | 0.65 | 2690 | 0.02 | 0.58 | 2688 | 0.54 | 0.32 | 1280 |
| ICES–bottom | 0.04 | 0.72 | 932 | 0.08 | 0.65 | 933 | 0.68 | 0.31 | 355 |
| ESA–CCI M1-3+10-12 | - | - | - | - | - | - | 0.10 | 0.75 | 8542 |
| ESA–CCI M4-9 | - | - | - | - | - | - | 0.79 | 0.81 | 8502 |
| ESA–CCI M4-6 | - | - | - | - | - | - | 1.19 | 0.79 | 8445 |
| ESA–CCI M7-9 | - | - | - | - | - | - | 0.39 | 0.78 | 8408 |
| ESA–CCI M1-12 | - | - | - | - | - | - | 0.43 | 0.81 | 8515 |

14,a,b). In the offshore areas, the estimates of the acclimative model are higher than those of the fixed model at the surface (Fig. 14,a), but slightly lower when water column averages are considered (Fig. 14,b), indicating that the phytoplankton growth occurs mostly at the bottom layers in the fixed–trait model, which is consistent with the daily vertical profiles in the fixed–trait model (Fig. B4c). Importantly, these results suggest that, a coastal gradient in phytoplankton concentrations is predicted by a non-acclimative model, which is presumably driven by the nutrient gradients (Fig. 11), but a much stronger gradient emerges when the acclimation processes are resolved. Specific to this example, towards the coast, phytoplankton adapt to the deteriorating light climate (Fig. B1) and increasing nutrient availability by investing more to the light harvesting machinery, as indicated by the increasing Chl:C ratios (Fig. 13), and thereby achieve higher coastal production rates than the case their physiology is fixed. As a result of increasing Chl:C ratios towards the coast, the chlorophyll concentrations display even a stronger gradient than that of the biomass: at the surface layer, increase of biomass concentrations towards the coast is about 3.5 fold (from about 10 to 35 mmmolC/m$^3$), while that of the chlorophyll is about 7 fold (from about 2 to 14 mg/m$^3$) along the transect shown in Fig. 14).

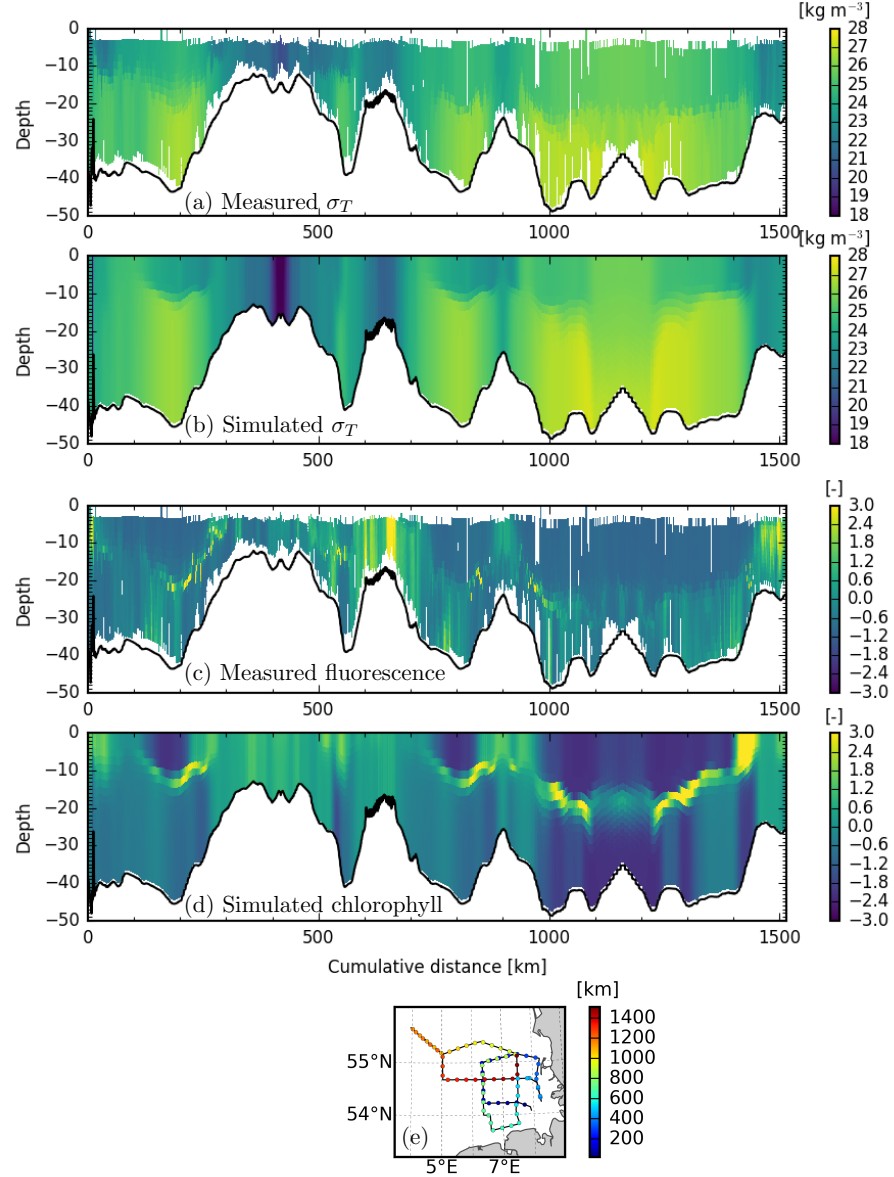

**Figure 9.** (a,b) $\sigma_T$, measured by Scanfish and estimated by the model (c,d) normalized anomalies of fluorescence measured by Scanfish and chlorophyll concentrations estimated by the model. Track of the cruise, which took place between 13-19 July 2010 is shown in (e).

## 4 Discussion

In order to assess the performance of the new model system presented for the first time in this study, we employed several independent observation sources and types: FerryBox measurements to assess the horizontal distribution of salinity (Fig. 3); sparse in-situ measurements from the ICES dataset for an overall evaluation of the physical and biogeochemical model (Fig.

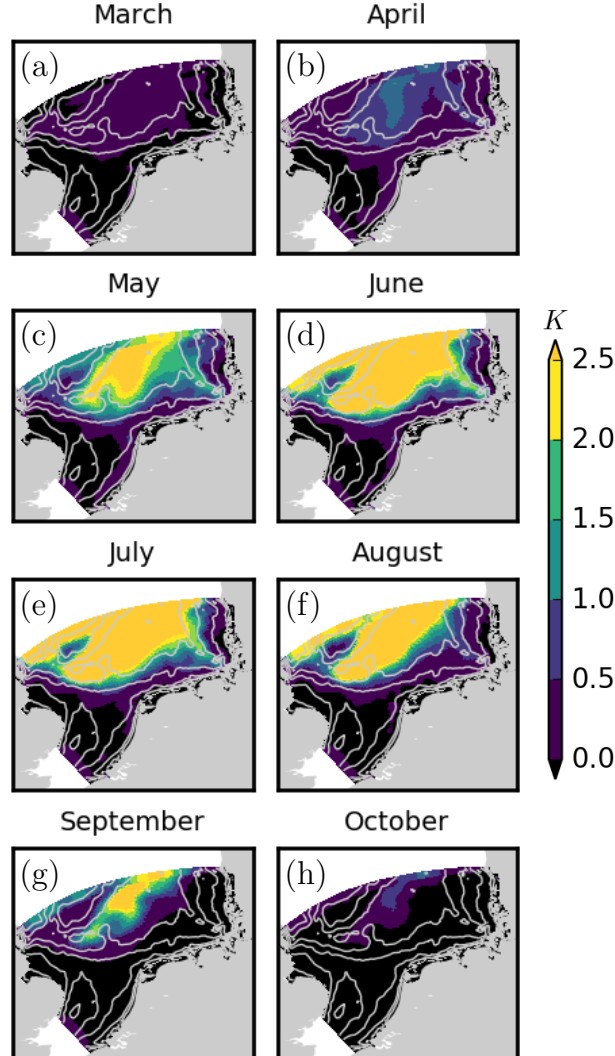

**Figure 10.** Average temperature difference [K] between the surface and bottom layers, averaged throughout 2000-2010 for each month. Gray lines show the isobaths.

4, Fig. 8); measurements from 19 monitoring stations for evaluating the estimates for DIN, DIP and chlorophyll at specific locations (Fig. 5-7); Scanfish measurements for evaluating the vertical density and chlorophyll profiles (Fig. 9); and finally the satellite observations for evaluating the model skill regarding the horizontal distribution of climatological chlorophyll concentrations (Fig. 12) and attenuation of light (Fig. B1).

5    The physical model can provide a realistic description of the hydrodynamical processes foremost relevant for modeling the biogeochemistry of the system. Horizontal circulation patterns are captured as evidenced by the salinity distribution being in

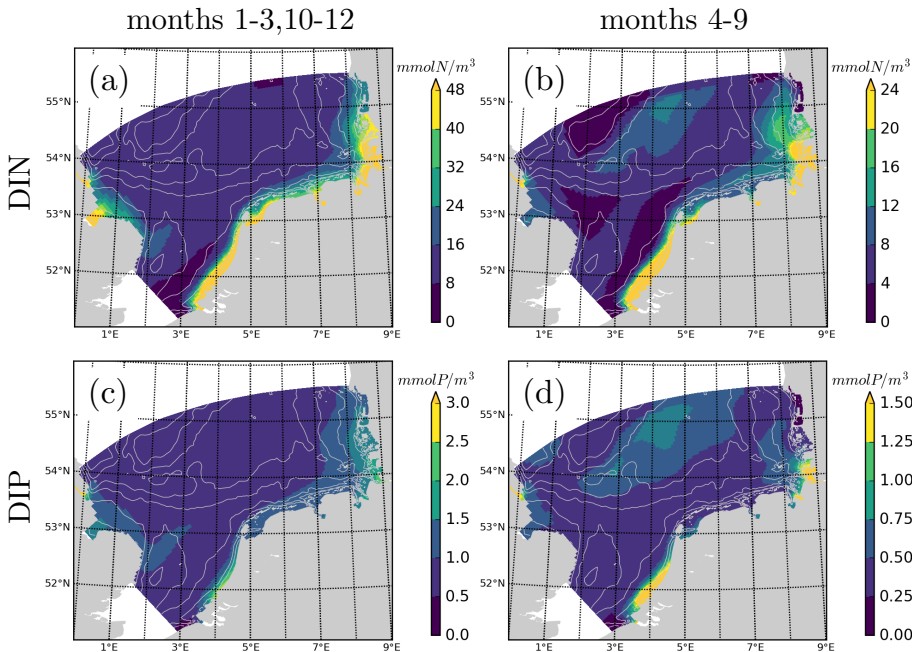

**Figure 11.** DIN (a,b) and DIP (c,d) concentrations at the surface layer, averaged over the non-growing (months 1-3 and 10-12, left) and growing seasons (months 4-9, right) for the entire simulation period (2000-2010). Concentrations at the bottom layer are almost identical for the months 1-3 and 10-12 and similar for the months 4-9. Gray lines show the isobaths. Note different color scales used for each panel, and that the scale used for DIN is 16 times that of DIP, such that identical coloring for DIN and DIP for the same season indicate a Redfield ratio of 16:1.

agreement with the observations (Fig. 3,4). Density structure of the system during summer, driven by a complex interplay between the salinity gradients, heat fluxes at the surface and tidal stirring is realistically captured, although the pycnocline depth seems to be underestimated (Fig. 9). Accordingly, temperature estimations match well with the observations, although there are cases where the stratification events are not reproduced by the model (Fig. 4), most of which are found to be within
5   the western portion of the model domain. The areal extent and seasonality of stratification (Fig. 10) is in agreement with those reported by earlier studies (Schrum et al., 2003; van Leeuwen et al., 2015). For nutrient concentrations relative bias of $\leq 12\%$ and correlation coefficients between 0.58-0.72 correspond to a high and moderate-high model skill, respectively (Table 1). For the pointwise comparisons of chlorophyll, model skill was moderate for the sparse measurements included in the ICES database, and for the stations in the German Bight (Table 1) but mostly low for the stations within the Western portion of the
10   model domain. Comparison of climatological averages of the simulated chlorophyll with those of the satellite observations resulted in high correlations for all seasons, and low to moderate-low bias, except during the early growing season (Table 1.

The model captures the subsurface chlorophyll maxima occuring in the deeper parts of the model domain (Fig. 9). The phenomena has been previously documented in the southern North Sea (Weston et al., 2005; Fernand et al., 2013). Former 3-D modeling studies, such as that of van Leeuwen et al. (2013), apart from capturing the presence of a deep chlorophyll maximum,

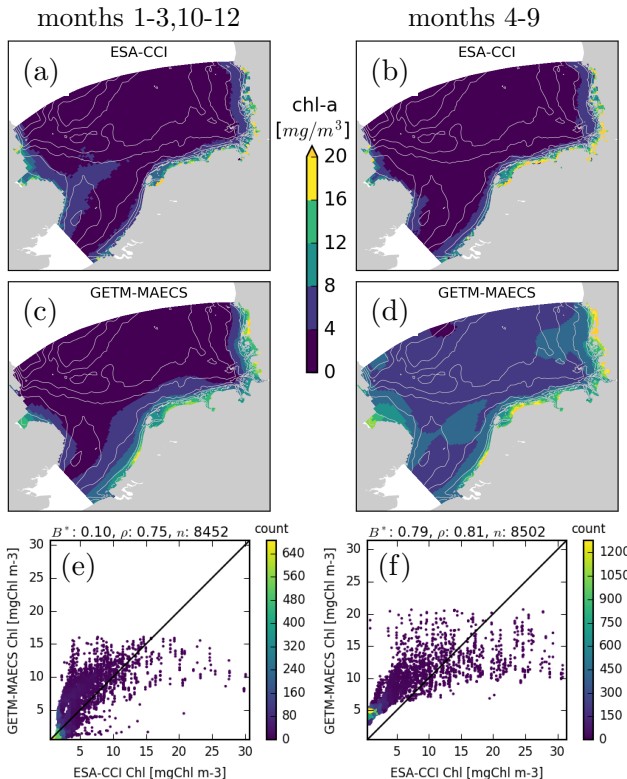

**Figure 12.** Comparison of satellite (ESA-CCI, a,b) and MAECS (c,d) estimates of surface chlorophyll concentrations averaged over 2008-2010 and for the non-growing (months 1-3 and 10-12, left) and growing seasons (months 4-9, right). 2-D histograms (e,f) show the number of occurrence of simulation-satellite data pairs. Gray lines in a-d show the isobaths. Normalized bias ($B^*$), Pearson correlation coefficients ($\rho$), and corresponding number of data points ($n$) are shown on top of scatter plots.

did not reproduce the rich variability revealed by the observations. Our comparative analysis shows that the formation and maintenance of such structures are critically dependent on the parametrization of the sinking rate of phytoplankton (Fig. B4a) and underwater light climate (Fig. B4b). Sinking speed of phytoplankton in the MAECS is inversely related to the nutrient quota of the cells, which mimics the internal buoyancy regulation ability of algae depending on internal nutrient reserves
5  (see Appendix A1) but also indirectly emulates chemotactic migration as typical for dinoflagellates (Durham and Stocker, 2012). This quota dependency of sinking results in considerable spatial variability, and significant differences between different seasons of the year (Fig. B3). The critical dependence of the formation and maintenance of vertical chlorophyll structures on the functional representation of sinking underlines the relevance of a consistent description of the intracellular regulation of nutrient storages. The latter, in turn, is determined by the metabolic needs, such as the intensity of light limitation, hence,
10  investments to the synthesis of pigmentory material (Wirtz and Kerimoglu, 2016). Indeed, the non-acclimative (fixed-trait) version of the model (Appendix B3) predicts qualitatively different vertical profiles of phytoplankton biomass (Fig. B4c), although the sinking parameterization in that simplified version is identical to that in the fully acclimative version. The non-

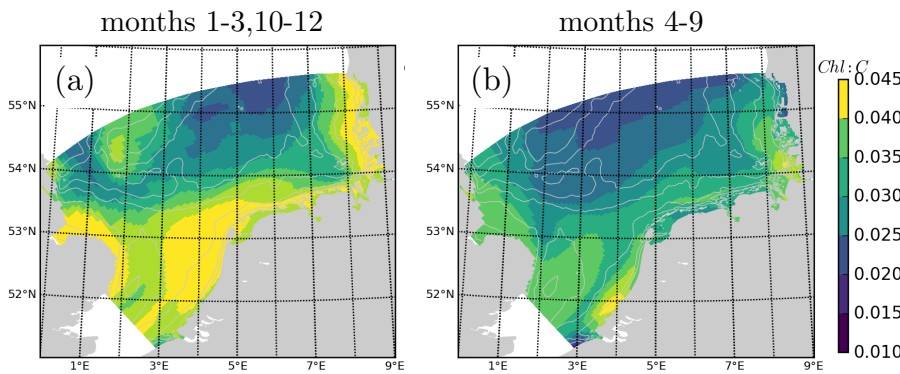

**Figure 13.** Chlorophyll:C ratio in phytoplankton, averaged over the non-growing (a) and growing season (b) of 2010. Gray lines in a-d show the isobaths.

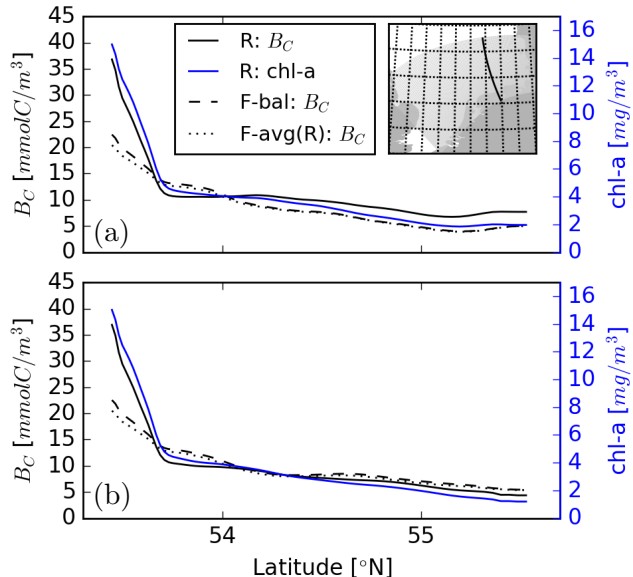

**Figure 14.** Annual average phytoplankton carbon (and for R, chlorophyll) concentrations in 2010; (a) at the surface layer; (b) averaged over the water column; obtained with R: reference (acclimative) model, F-bal: fixed-physiology model with balanced investments, F-avg(R): fixed-physiology model with allocation parameters as average trait values produced by R.

acclimative model version might be tuned to match the observed vertical distributions of phytoplankton, however, this would probably be at the cost of compromised performance in some other respects, such as the horizontal gradients, or timing and amplitude of chlorophyll blooms.

We conclude from the extensive model performance assessment that the model reproduces the main physical and biogeo-
5   chemical characteristics of the southern North Sea especially within the German Bight, where the model resolution is finest

(Fig. 1), and the influence of fluxes at the open boundaries is relatively small, given the predominantly counter-clockwise circulation pattern (Becker et al., 1992). The process of performance assessment also helped identifying the possibilities for further model refinement. For instance, a comparison with the satellite observations revealed that the light attenuation in the offshore areas is overestimated by the model, primarily because of the contributions by the climatological SPM forcing (Fig.
B1). A likely consequence of the overestimated attenuation is an underestimation of the depth of primary production (e.g., Fig.B4b), and this may, in turn, explain the overestimated chlorophyll concentrations in the offshore areas during the growing season (Fig. 12b,d). Another source of error regarding the SPM-caused turbidity is that, at specific coastal sites, like at the Noordwijk-10 station, the measured suspended particulate material (SPM) concentrations show considerable inter-annual variations that can obviously be not represented by the climatological SPM forcing (Fig. B2), which may explain the partic-
ularly low correlation coefficient (0.14) obtained at this station for chlorophyll. A better representation of the SPM-caused turbidity might be achieved by an explicit description of the SPM dynamics (e.g., as in van der Molen et al., 2016). Coupling the biogeochemical model with such an SPM model would then also allow the description of the two-way interactions, i.e., not only light limitation (Tian et al., 2009), but also by the acceleration of sinking of SPM by the production of transparent exopolymer particles (Schartau et al., 2007; Maerz et al., 2016). At the stations within the ROFIs of major rivers, such as the
Norderelbe and S. Amrum (Fig. 5) and Noordwijk–2&–10 (Fig. 6), the skill scores are relatively low (Table 1). These stations, especially the Noordwijk–2 and –10 are located where the concentrations change dramatically within 10-15 km (e.g, Fig. 11). Accordingly, a slight error by the physical model in predicting the salinity front, e.g., because of an inadequate representation of the tidal dynamics, might result in considerable deviation of the estimated concentration of biogeochemical variables from the measurements. Relatively coarser model resolution at around the Dutch coast might therefore explain the consistently lower
skill scores obtained at the Dutch stations. Identifying such potential inadequacies of the physical model requires further investigation, such as an assessment of the tidal constituents at the tidal gauges (e.g., Gräwe et al., 2016). Another potential source of error for the mismatches within the ROFIs is the potential flaws in the description of riverine loadings, such as assuming that the non-dissolved fractions of the total nitrogen and total phosphorus being entirely in labile form (Sect. 2.2.3). Although the earlier replenishment of phosphorus relative to nitrogen in the coastal sites is often reproduced (e.g., Sylt, Noordwijk-2,
Noordwijk-10, Terschelling-4), some delays occur in stations like Norderney, which probably reflects the oversimplification of the benthic processes with respect to the description of oxygen-driven iron-phosphorus complexation kinetics (Appendix A2), which has been suggested to be the main driver for the phenomena in the coastal areas (Jensen et al., 1995; van Beusekom et al., 1999; Grunwald et al., 2010).

The model predicts steep coastal gradients in nutrient concentrations (Fig. 11), in line with observations (e.g., Brockmann
et al., 1999; Hydes et al., 2004). Maintenance of these gradients during winter is explained by the limited horizontal mixing due to the density gradients caused by the freshwater influx from the land (Simpson et al., 1993; Hydes et al., 2004), and trapping of this nutrient-rich freshwater at the coast due to the along-shore currents in the study system driven by predominantly westerly winds and the coriolis forcing (Becker et al., 1992; Simpson et al., 1993). During the warmer seasons when the offshore waters are stratified, owed to the presence of horizontal salinity gradients, a mechanism similar to the estuarine circulation (Simpson
et al., 1990) was suggested to further promote these gradients along the Wadden Sea, also in regions far from river inputs

(Burchard et al., 2008; Flöser et al., 2011; Hofmeister et al., 2016). Coastal waters remaining to be nutrient-replete during the growing season lead to high phytoplankton concentrations (Fig. 12) despite the higher turbidities at the coastal waters (Fig. B1).

Comparison of the present model with earlier attempts is neither in the scope of this study, nor is possible without a dedicated benchmarking effort, using standardized forcing data and skill performance assessment datasets and methodology (e.g., as in Friedrichs et al., 2007). Even a qualitative comparison is difficult, given that spatial and temporal binning of the data, frequently employed in model validation (like in our Fig. 12), can dramatically impact the skill scores, and that pointwise comparisons with sparse observation datasets (like in our Fig. 4 and 8) are rarely performed (de Mora et al., 2013). However, skill of the presented model in estimating the chlorophyll concentrations in the SNS can argued to be at least comparable to those of the recent modelling applications for a relevant region (e.g., Edwards et al., 2012; de Mora et al., 2013; Ciavatta et al., 2016; Ford et al., 2017, noting that all these studies had larger model domains, and were evaluated for different time intervals). This is noteworthy, given that phytoplankton is represented by a single species in our model, whereas in other modelling approaches, several species or groups are resolved. Inclusion of multiple functional types is motivated by the spatial and seasonal variability in the phytoplankton composition observed in the field: coastal areas of the SNS are dominated by diatoms, throughout the year in some sites (e.g., Alvarez-Fernandez and Riegman, 2014), and during spring in some others, later to be replaced by *Phaeocystis* during summer (e.g., van Beusekom et al., 2009), whereas the offshore areas are often dominated by dinoflagellates especially during summer (Freund et al., 2012; Wollschläger et al., 2015). These phytoplankton groups differ from each other by a number of traits, including the physiological traits that determine their ability to access the (mineral and light) resources and build biomass. For instance, in an experimental work, two diatom species were shown to have on average more than three fold higher Chl:C ratios than those of two dinoflagellate species (Chan, 1980), making them therefore more tolerant to the light-limited conditions of the turbid, coastal waters. In the presented approach, the cellular composition of the single, but acclimative phytoplankton group dynamically approaches towards (for some traits, instantaneously adopt) the physiological state of the ideal resource competitor in a given environment, which, in nature, happens through various processes from the plastic response of the individual cells to the species sorting at the community level. In a traditional, functional plankton type (PFT) model on the other hand, the species with most suitable traits will become the most dominant among others, while the proximity of the physiological traits to the theoretical optima, thus the overall productivity will be determined by the resolution of physiological traits as represented by the defined clones. The worst case is when there is only one, non-acclimative group, as illustrated in our experiment: at the turbid but nutrient–rich coastal areas, prioritization of the light harvesting over the nutrient acquisition machinery, as evidenced by the higher Chl:C ratios predicted by the acclimative model (Fig. 13), leads to better fitness, thus higher phytoplankton concentrations in comparison to the non-acclimative equivalents (Fig. 14). Moreover, because of the high Chl:C ratios at the coastal areas (Fig. 13), chlorophyll concentrations display even steeper gradients than the phytoplankton concentrations (Fig. 14). The transitional Chl:C pattern suggested by our model has been previously identified based on monitoring data by Alvarez-Fernandez and Riegman (2014). The Chl:C ratios ranging between 0.01-0.1 gChl/gC at the coastal stations and 0.002-0.02 gChl/gC at the offshore stations reported by Alvarez-Fernandez and Riegman (2014) envelope our estimated seasonal average values of 0.045 and 0.015 within the respective regions. According to the simulation

results, Chl:C ratios differ considerably also between the non-growing and growing season, with higher values during winter, due to low light availability. A similar seasonal amplitude in Chl:C has been found by Llewellyn et al. (2005) for the English Channel with higher ratios during winter.

As mentioned above, physiological composition is not the only relevant trait for determining the community composition in the study system. Diatoms are fast growers and defended against the efficient microzooplankton grazers, but this comes at the cost of silicate requirement for their growth (Loebl et al., 2009) and higher sedimentation losses (Riegman et al., 1993). *Phaeocystis* are slow growers, but by forming large colonies, they are well defended against zooplankton (Peperzak et al., 1998). Finally, the dinoflagellates, also despite being slow growers, are mobile (Durham and Stocker, 2012) and mostly have access to alternative nutrient sources through their phagotrophic abilities (Löder et al., 2012). Representation of zooplankton with a single group may also be an oversimplification, as the microzooplankton and mesozooplankton have considerably different growth rates (Hansen et al., 1997) and functional responses to prey availabilities (Kiørboe, 2011). Moreover, effects of temperature on mesozooplankton occurs through phenological shifts (e.g., Greve et al., 2004) that might have a determining role on the maximum chlorophyll concentrations (van Beusekom et al., 2009), which can probably be only partially reflected by the simple Q10 rule we applied for grazing rates (Appendix A1). None of these ecophysiological aspects were taken into account in our model, and this may explain some of the discrepancies between the simulated and observed chlorophyll concentrations. In future work, inclusion of few other phytoplankton groups –each being acclimative, and one additional zooplankton group is foreseen. While the consideration of other phytoplankton traits should be straightforward, inclusion of phagotrophy as an additional physiological allocation trait represented by a state variable is possible (e.g., as in Chakraborty et al., 2017), but would require re-derivation of the model equations.

## 5  Conclusions

In this study, we described the implementation of a coupled physical-biogeochemical model to the Southern North Sea (SNS) and analyzed the model results in comparison to a large collection of *in-situ* and remote sensing data. The model system accounts for key coastal processes, such as the forcing by local atmospheric conditions, riverine loadings of inorganic and organic material, atmospheric nitrogen deposition, spatio-temporal variations in the underwater light climate, major benthic processes and nutrient concentrations at open boundaries, and importantly, it hosts a novel model of phytoplankton growth, which replaces otherwise heuristic formulations of photosynthesis and nutrient uptake with mechanistically sound ones (Wirtz and Kerimoglu, 2016). Based on comparisons with a number of data sources, we conclude that the model system can produce a realistic decadal hindcast of the German Bight for the period 2000-2010, in terms of both the temporal and spatial distribution of key ecosystem variables, as well as a large area of validity, i.e., both in coastal and offshore regions of the German Bight.

In 3-D model applications so far, photoacclimation of phytoplankton has been either ignored altogether, or it was accounted for in a heuristic sense, where the change in Chl:C ratio is described based on an empirical relationships (Blackford et al., 2004; Fennel et al., 2006). In our model, adaptation of the phytoplankton community to the light and nutrient environment is represented by dynamically changing and instantaneously optimized trait values as described extensively by Wirtz and

Kerimoglu (2016). Our findings suggest that the steep chlorophyll gradients across the coastal transition zone is mainly driven by the nutrient gradients, but amplified first by the acclimative capacity, then further by higher Chl:C ratios at the coastal waters. The large variations in simulated Chl:C ratios within the SNS, both in a space and time, indicate that ignoring photoacclimation can lead to potentially flawed estimates for primary production or phytoplankton biomass as was recently pointed out by Arteaga et al. (2014) and Behrenfeld et al. (2015), based on the variability of Chl:C ratios at global scales. Here we show that this warning applies especially in the coastal environments characterized by steep resource gradients, which may be critical, given the increasing recognition of the role of coastal-shelf systems in the global carbon and nutrient cycling (Fennel, 2010; Bauer et al., 2013).

## Appendix A: Detailed Model Description

### A1 Pelagic Module

Local source-sink terms for all dynamic variables, functional description of processes and relationships between quantities and parameters used for the pelagic module are provided in Table A1–A3.

Importantly, the biogeochemical model resolves photoacclimation of phytoplankton, described by dynamical partitioning of resources to light harvesting pigments (Eq. A7), enzymes involved in carboxylation reactions (Eq. A8) and nutrient uptake sites (i.e., $f_{LH} + f_C + f_V = 1$) as in Wirtz and Pahlow (2010). Uptake of each nutrient is optimally regulated (as expressed by $a_i$ in Eq. A16-A17), and following Pahlow (2005); Smith et al. (2009), optimality along the affinity-intracellular transport trade-off ($A_i = f_i^A \cdot A_i^*$ and $V_{max,i} = (1 - f_i^A) \cdot V_{max,i}^*$, see Table A3 for the definition of parameters). As a second novelty, the growth model uniquely describes the interdependence between limiting nutrients to be variable between full inter-dependence (as in product rule) and no-interdependence (as in Liebig's law of minimum) as a function of nitrogen quota (See Eq. A13). For a detailed explanation of the phytoplankton growth model and solution of differential expressions in Eq. A7,A8 and A17 refer to Wirtz and Kerimoglu (2016). For enabling the spatial transport of the 'property variables' of phytoplankton such as $Q_i$, $f_{LH}$ and $f_{LH}$, they have been transformed to bulk variables by multiplying with the phytoplankton carbon biomass, i.e., $B_C$. Parameterization of the phytoplankton model, except $\theta_C$, fall within the range of parameter values used by Wirtz and Kerimoglu (2016). The exact values of the parameters were established by manual tuning, given that important phytoplankton species such as various diatom and dinoflagellate species, and *Phaeocystis* sp. that dominate the phytoplankton composition in the SNS (eg., Wiltshire et al., 2010) have not been studied formerly within the presented model framework.

Phytoplankton losses are due to aggregation and zooplankton grazing (see below). Specific aggregation loss rate (Eq. A19) is described as a function of DOC that mimics transparent exopolymer particles (Schartau et al., 2007) to account for particle stickiness, multiplied by the sum of phytoplankton biomass and of POM reflecting density dependent interaction, which is equivalent to a quadratic loss term. Zooplankton dynamics are described only in terms of their carbon content, assuming stoichiometric homeostasis (Sterner and Elser, 2002). Grazing is described by a Holling Type-3 function of prey concentration (Eq. A20). A lumped loss term accounts for the respiratory losses and exudation of N and P in dissolved inorganic form (Eq. A21), which are adjusted depending on the balance between the stoichiometry of zooplankton and that of the ingested food

for maintaining the homeostasis (Eq. A22). The effect of organisms at higher trophic levels, mainly by fish and gelatinous zooplankton are mimicked by a density-dependent mortality of zooplankton, modified by a function of total attenuation of Photosynthetically Available Radiation (PAR) (Eq. A23) to account for higher predation pressure exerted by fish at the offshore regions of the North Sea, which amounts to about two times that in the coastal regions according to the estimates based

on trawl surveys (Maar et al., 2014). All kinetic rates were modified for ambient water temperature, T (K) using the Q10 rule parameterized specifically for autotrophs and small heterotrophs (=bacteria for hydrolysis and remineralization) and for zooplankton.

**Table A1.** Source-sink terms of the dynamic variables of the pelagic module. The index $i$ represents the elements C, N, P. By definition, $Q_C = Q_C^Z = 1$, $Q_i = B_i / B_C$. Dynamics of Dissolved Inorganic Carbon (DIC) is not resolved, thus (Eq. A4) not integrated for $i$=C. Description of processes or functional relationships (capital letters) and of parameters (small letters) are provided in Tables A2 and A3, respectively.

| | | | |
|---|---|---|---|
| Autotrophic biomass | $s(B_C)$ | $= (V_C - \sum_i V_i \zeta_i - L_A) \cdot B_C - G \cdot Z_C$ | (A1) |
| Internal quota | $s(Q_i)$ | $= V_i - V_C Q_i$ | (A2) |
| Zooplankton | $s(Z_C)$ | $= (\gamma G - M - L_Z) \cdot Z_C$ | (A3) |
| Dissolved inorganics | $s(\mathrm{DIM}_i)$ | $= L_Z Z_C Q_i^Z + r_{\mathrm{DOM}} \, \mathrm{DOM}_i - V_i \, B_C$ | (A4) |
| Dissolved organics | $s(\mathrm{DOM}_i)$ | $= r_{\mathrm{POM}} \, \mathrm{POM}_i - r_{\mathrm{DOM}} \mathrm{DOM}_i$ | (A5) |
| Particulate organics | $s(\mathrm{POM}_i)$ | $= L_A B_C Q_i + (1-\gamma) G Z_C Q_i + M Z_C Q_i^Z - r_{\mathrm{POM}} \mathrm{POM}_i$ | (A6) |
| Carboxylation (Rub) | $s(f_C)$ | $= \delta_C \cdot \left( \frac{\partial V_C}{\partial f_C} + \sum_i \frac{\partial V_C}{Q_i} \frac{\mathrm{d}Q_i}{\mathrm{d}f_C} \right)$ | (A7) |
| Pigmentation (Chl) | $s(f_{\mathrm{LH}})$ | $= \delta_{\mathrm{LH}} \cdot \left( \frac{\partial V_C}{\partial f_{\mathrm{LH}}} + \sum_i \frac{\partial V_C}{\partial Q_i} \frac{\mathrm{d}Q_i}{\mathrm{d}f_{\mathrm{LH}}} \right)$ | (A8) |

## A2  Benthic Module

The benthic module provides simplistic descriptions of the degradation of N and P from POM to DIM, their fluxes across the

benthic-pelagic interface, removal of N due to denitrification and accounts for the sorption dynamics of P.

POM degrades into DIM in one step, described as a first order reaction, the rate of which is modified for temperature using the Q10 rule. POM flux into the sediments by settling of material from the water fuels the benthic POM (bPOM) (Eq.A4). On the other hand, diffusive flux of DIM is possibly bi-directional, depending on the concentration gradient between water and soil (Eq. A4). Inorganic phosphorus (denoted as TIP,(Eq. A4) is assumed to exist in two states: sorbed and dissolved state.

Fraction of the sorbed state is given by a function of dissolved oxygen (DO), to account for the production and adsorption of Fe-P complexes in oxic conditions and their desorption at anoxic conditions (Eq. A4). Given the observed inverse relationship between temperature and oxygen concentrations in sediments (e.g., Jensen et al., 1995), DO is heuristically estimated as a function of temperature ($T$) to capture the seasonal hypoxia events. Resulting functional relationships between the sorbed fraction of TIP, T and DO are shown in Fig. A1(a-b). Following the simplistic approach used for the ECOHAM model (Pätsch

and Kühn, 2008), denitrification rate is estimated from the degradation rate (Eq. A4) using empirically derived ratios and

**Table A2.** Process descriptions and functional relationships. The index $i$ represents the elements C, N and P. The index $j$ represents groups with different Q10 values. Description of parameters (small letters) are provided in Table A3.

| | | | |
|---|---|---|---|
| Carbon uptake | $V_C$ | $= P \cdot g_n\left(\mathcal{C}_n\left(q_N, \mathcal{C}_n(q_P, q_C)\right)\right) - \sum_i \zeta_i V_i$ | (A9) |
| Light lim. primary prod. | $P$ | $= f_C\, P_{\max} \cdot \left(1 - e^{-\alpha\,\theta PAR/P_{\max}}\right)$ | (A10) |
| Chlorophyll conc. in chloroplasts | $\theta$ | $= \theta_C \dfrac{f_{LH}}{q_N f_C}$ | (A11) |
| Relative resource availability | $q_i$ | $= \dfrac{Q_i - Q_i^0}{Q_i^* - Q_i^0}$ | (A12) |
| Co-limitation function | $\mathcal{C}_n(q_i, q_j)$ | $= q_i \cdot g_n\left(\dfrac{q_j}{q_i}\right) \cdot \left(1 + \dfrac{q_i q_j}{n} + log(4^{-1/n} + 0.5/n)\right)$ | (A13) |
| Queuing function | $g_n(r)$ | $= \dfrac{r - r^{1+n}}{1 - r^{1+n}}$ | (A14) |
| Degree of independence | $n$ | $= n^* \cdot (1 + q_N)$ | (A15) |
| Nutrient uptake | $V_i$ | $= f_V\, a_i \cdot \left(V_{\max,i}^{-1} + (A_i DIM_i)^{-1}\right)^{-1}$ | (A16) |
| Uptake activity | $a_i$ | $= \left(1 + e^{-\tau_v \frac{dV_C}{da_i}}\right)^{-1}$ | (A17) |
| Flexibility ($X =$C, LH) | $\delta_X$ | $= f_X \cdot (1 - f_X)$ | (A18) |
| Losses due to aggregation | $L_A$ | $= L_A^* \cdot \left(\dfrac{a_{DOC} DOM_C}{1 + a_{DOC} DOM_C}\right) \cdot \left(B_N + POM_N\right)$ | (A19) |
| Grazing | $G$ | $= G_{max} \dfrac{B_C^2}{K_G^2 + B_C^2}$ | (A20) |
| Zooplankton loss | $L_Z$ | $= m_r Q_i^Z - S + \max(0, \gamma G(Q_i - Q_i^Z))$ | (A21) |
| Zooplankton homeostatic adjustment | $S$ | $= $ if $(m_r Q_i^Z + \gamma G(Q_i - Q_i^Z)) < 0$: $(1-\gamma)GQ_i$; else:0 | (A22) |
| Zooplankton mortality | $M_i$ | $= m_f \cdot \left(1 + \Delta_f \cdot \left(1 - \left(1 + e^{s_k \cdot (k_{tot}^* - k_{tot})}\right)^{-1}\right)\right) \cdot Z_C$ | (A23) |
| Total PAR attenuation | $k_{tot}$ | $= -\dfrac{z}{\eta_2} - \int_z^0 \sum_i k_{c,i} c_i(z')dz'$ | (A24) |
| Phytoplankton sinking | $w_B$ | $= w_B^0 + w_B^* e^{-s_w q_N q_P}$ | (A25) |
| Temperature dependence | $F_T^j$ | $= Q10_j^{(T - T_{ref}/10)}$ | (A26) |

stoichiometric conversions, considering in addition the limitation imposed by the available DIN and inhibition by DO (Soetaert et al., 1996). Resulting functional relationships between denitrification, T and DO are shown in Fig. A1(c-d).

**Table A3.** Parameters of the pelagic module. Codes for sources: c: calibrated; a: assumed; l: typical literature value; d: by definition; 1:Wirtz and Kerimoglu (2016); 2:Hansen et al. (1997, for copepods); 3:Oubelkheir et al. (2005); 4:Stedmon et al. (2001); 5:Maar et al. (2014)

| Symbol | Description | Value | Unit | Source |
|---|---|---|---|---|
| Parameters relevant to phytoplankton | | | | |
| $\alpha$ | Light absorption coefficient | 0.2 | $m^2$ mmolC($\mu$E gCHL)$^{-1}$ | 1,c |
| $A_P^*$ | Affinity to PO$_4$ | 0.15 | $m^3$(mmolC d)$^{-1}$ | 1,c |
| $A_N^*$ | Affinity to inorganic N | 0.4 | $m^3$(mmolC d)$^{-1}$ | 1 |
| $P_{max}^*$ | Potential photosynthesis rate | 9.0 | d$^{-1}$ | 1,c |
| $\theta_C$ | CHL-a/C ratio in chloroplasts | 1.0 | gChl molC$^{-1}$ | c |
| $Q_N^0$ | Subsistence quota for N | 0.035 | molN molC$^{-1}$ | 1,c |
| $Q_P^0$ | Subsistence quota for P | 0.0 | molP molC$^{-1}$ | 1 |
| $Q_N^*$ | Reference N quota | 0.17 | molN molC$^{-1}$ | 1,c |
| $Q_P^*$ | Reference P quota | 0.0055 | molP molC$^{-1}$ | 1 |
| $n^*$ | specific independence | 4.0 | - | 1,c |
| $V_{max,N}^0$ | Potential N uptake rate | 1.0 | molN (mmolC d)$^{-1}$ | 1,c |
| $V_{max,P}^0$ | Potential P uptake rate | 0.1 | molP (mmolC d)$^{-1}$ | 1,c |
| $\zeta_N$ | C cost of N assimilation | 4.0 | molC molN$^{-1}$ | 1,c |
| $\zeta_P$ | C cost of P assimilation | 24.0 | molC molP$^{-1}$ | 1,c |
| $\tau_v$ | Relaxation time scale for $a_i$ | 10 | d | 1 |
| $w_B^*$ | Maximum quota-dependent sinking rate | 3.0 | m d$^{-1}$ | c |
| $w_B^0$ | Background sinking rate | 0.2 | m d$^{-1}$ | c |
| $s_w$ | Scaling coefficient for sinking function | 4.0 | - | c |
| $L_A^*$ | Maximum aggregation rate | 0.003 | molC molN$^{-1}$ | c |
| $a_{DOC}$ | DOC specific aggregation coefficient | 0.1 | mmolC m$^{-3}$ | c |
| $Q10_B$ | Q10 coefficient for autotrophs and bacteria | 1.5 | - | 1 |

(continued on the next page)

| Symbol | Description | Value | Unit | Source |
|--------|-------------|-------|------|--------|
| **Parameters relevant to zooplankton** | | | | |
| $Q_N^Z$ | N:C ratio | 0.25 | molN molC$^{-1}$ | 1 |
| $Q_P^Z$ | P:C ratio | 0.02 | molP molC$^{-1}$ | 1 |
| $G_{max}$ | Max. grazing rate | 1.2 | d$^{-1}$ | 2 |
| $\gamma$ | Assimilation efficiency | 0.35 | - | 2 |
| $K_G$ | Half saturation constant for grazing | 20.0 | mmolC m$^{-3}$ | 2 |
| $m_r$ | Basal respiration rate | 0.02 | d$^{-1}$ | 1 |
| $m_f$ | Base mortality rate | 0.02 | m$^3$ (mmolC d)$^{-1}$ | c |
| $\Delta_f$ | Maximum incremental mortality factor | 1.0 | - | 5 |
| $k_{tot}^*$ | Critical total PAR attenuation | 0.4 | m$^2$ mmolC$^{-1}$ | c |
| $s_k$ | Scaling coefficient for mortality function | 10.0 | mmolC m$^{-2}$ | c |
| $Q10_Z$ | Q10 coefficient for zooplankton | 2.0 | - | 1 |
| **Other biogeochemical parameters** | | | | |
| $T_{\text{ref}}$ | Reference temperature for kinetic rates | 288 | K | d |
| $w_{POM}$ | Sinking rate of POM | 6.0 | m d$^{-1}$ | c |
| $r_{POM}$ | Hydrolysis rate | 0.03 | d$^{-1}$ | c |
| $r_{DOM}$ | Remineralization rate | 0.03 | d$^{-1}$ | c |
| $k_B$ | Attenuation coefficient for phytoplankton | 0.015 | m$^2$ mmolC$^{-1}$ | 3 |
| $k_{POC}$ | Attenuation coefficient for POC | 0.01 | m$^2$ mmolC$^{-1}$ | 3 |
| $k_{DOC}$ | Attenuation coefficient for DOC | 0.0025 | m$^2$ mmolC$^{-1}$ | 4 |

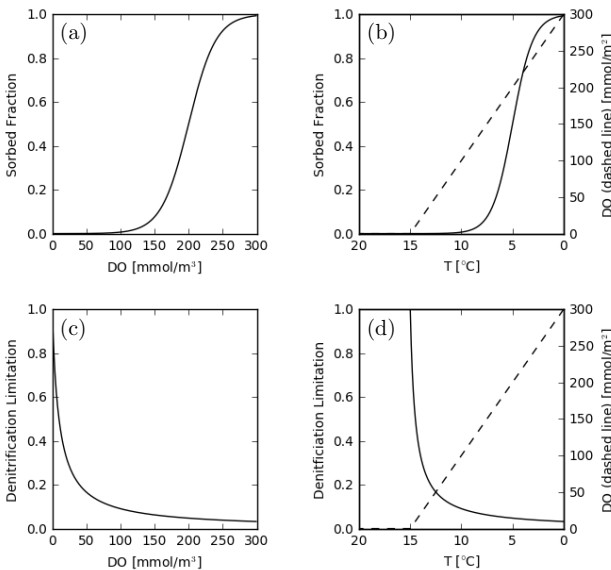

**Figure A1.** Fraction of sorbed fraction of benthic phosphorus as functions of DO (a) and T (b), regulation of benthic denitrification rate as functions of DO (c) and T (d), and DO as a function of T (b,d).

**Table A4.** Source-sink terms of the dynamic variables, and functional relationships of the benthic module. $sDIM_X = sDIX$ and $sPOM_X = sPOX$, where $X = \{N, P\}$. Description of parameters (small letters) are provided in TableA5

| Dynamics: | | |
|---|---|---|
| Benthic POM | $s(bPOM_i)$ | $= E_{POM} - R_i$ |
| Benthic TIP | $s(bTIP)$ | $= E_{DIP} + R_P$ |
| Benthic DIN | $s(bDIN)$ | $= E_{DIN} + R_N - \Upsilon$ |

| Functional relationships: | | | |
|---|---|---|---|
| Benthic Remineralization rate | $R_i$ | $= r_B \cdot sPOM_i$ | (A27) |
| POM exchange with water | $E_{POX}$ | $= \psi_{POM} * POM$ | (A28) |
| DIM exchange with water | $E_{DIX}$ | $= D_{DIM} \cdot \frac{DIM - bDIM}{\Delta_Z}$ | (A29) |
| Fraction of inorganic P in dissolved phase | $bDIP$ | $= 1 - bAP$ | (A30) |
| Fraction of inorganic P in adsorbed phase | $bAP$ | $= \left(1 + e^{sa \cdot (bDO^* - bDO)}\right)^{-1}$ | (A31) |
| Denitrification | $\Upsilon$ | $= c_{O:N} c_{N:O} R_N \cdot \left(\frac{bDIN}{K_{\Upsilon,DIN} + bDIN}\right) \cdot \left(1 - \frac{bDO}{K_{\Upsilon,DO} + bDO}\right)$ | (A32) |
| Benthic dissolved oxygen | $bDO$ | $= 300.0 - c_{DO} * T$ | (A33) |
| Temperature dependence | $F_T^b$ | $= Q10_b^{(T - T_{ref}/10.0)}$ | (A34) |

**Table A5.** Parameters of the benthic module. Codes for sources: c:calibrated; a:assumed; l: typical literature value; 1:Soetaert et al. (1996); 2:Seitzinger and Giblin (1996)

| Symbol | Description | Value | Unit | Source |
|---|---|---|---|---|
| $r_b$ | Benthic degradation rate | 0.05 | $d^{-1}$ | c |
| $\psi_{POM}$ | Sinking velocity of POM across the benthic-pelagic interface | 3.0 | $d^{-1}$ | c |
| $D_{DIM}$ | Diffusivity of DIM across the benthic-pelagic interface | 5e-4 | $m^2\ d^{-1}$ | 1 |
| $\Delta_Z$ | Thickness of the boundary layer | 0.2 | m | a |
| $c_{DO}$ | DO-T coefficient | 20 | mmolO $K^{-1}$ | c |
| $K_{\Upsilon,DO}$ | Half saturation for DO inhibition of denitrification | 10 | mmolO $m^{-2}$ | 1 |
| $K_{\Upsilon,DIN}$ | Half saturation for DIN limitation of denitrification | 30 | mmolO $m^{-2}$ | 1 |
| $c_{O:N}$ | Consumed oxygen per degraded nitrogen | 6.625 | molO $molN^{-3}$ | a |
| $c_{N:O}$ | Denitrified N per consumed oxygen | 0.116 | molN $molO^{-3}$ | 2 |
| $s_a$ | Scaling coefficient for DO-sorption relationship | 0.05 | $m^3$ $mmolC^{-1}$ | c |
| bDO$^*$ | Critical benthic DO concentration for P-soprtion | 200 | molO $m^{-3}$ | c |
| $Q10_b$ | Q10 coefficient for benthic reactions | 2.0 | - | 1 |

## Appendix B: Additional Analyses

### B1 Realism of Light Climate

Main driver of the spatio-temporal variations of the light climate in the southern North Sea is recognized to be the suspended particulate material (SPM) concentrations (e.g., Tian et al., 2009). In this study, light attenuation caused by SPM was provided as a 2-D forcing field of monthly climatologies (as required by GETM), which was extracted for 5 m depth from an original 3-D climatological (daily) dataset constructed by a statistical regression approach (Heath et al., 2002) and used in ECOHAM (e.g., Große et al., 2016). Here we aim to gain insight into the realism of the light climate represented by the model, in particular with respect to the turbidity caused by SPM, as the rest of the turbidity is mainly caused by the simulated variables, in particular phytoplankton, realism of which is discussed extensively in the main text (e.g., Fig. 8, 9, 12).

Along a transect following the model grid with a principal east-west axis, the qualitative pattern of the total attenuation estimated by the model at the surface layer agrees with that estimated by a satellite product, both averaged for 2008-2010 (Fig. B1), high towards the British coast at the western border, even higher at the German coast at the eastern border, and low in between. However, the variability in the attenuation estimated by the model is dampened than that by the satellite product, in particular, in the form of an overestimation of the offshore values. Some of this mismatch might be owed to the difference between the wavelengths at which the attenuation is provided by the model (average between 400-700 nm) and satellite product (490 nm), but this is not expected to be the major reason. Therefore, we conclude that the turbidity in the offshore regions is overestimated by the model. We further note that about 80% of the total attenuation at these offshore regions is due to SPM used as forcing (Fig. B1).

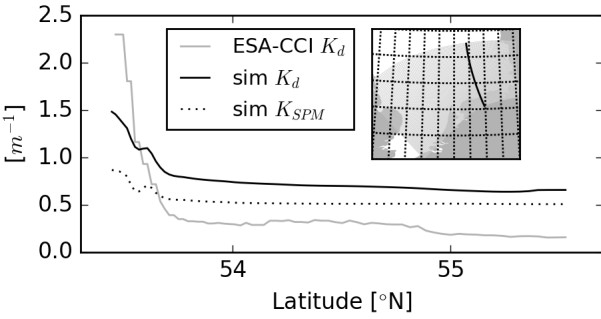

**Figure B1.** Total light attenuation along the transect shown in the inset as estimated by the satellite product (ESA-CCI $K_d$, solid gray line) and by the model at the surface layer (sim $K_d$, solid black line). Light attenuation caused only by SPM used as model forcing is shown separately (sim $K_{SPM}$, solid dotted line). All values represent averages between 2008-2010. The satellite estimates are from 490 nm wavelength and the model estimates represent the average within the PAR (400-700 nm) spectrum.

SPM concentrations, hence turbidity, within the regions located at the transition between the low-turbidity offshore waters to high-turbidity coastal waters (e.g., between 8-9 °E in Fig. B1,) display a considerable amount of sub-annual and inter-annual variability, driven by a combination of processes like riverine discharges, salinity fronts and sediment resuspension (e.g., Tian

et al., 2009; Su et al., 2015). This is exemplified by the SPM concentrations in 2008 and 2009 measured at the Noordwijk-10 station (Fig. B2). Such variations in SPM-caused turbidity are, by definition, not captured by the monthly climatology data, and in turn, leading to mismatches in the simulated phytoplankton, for instance in the form of timing errors (Fig.6).

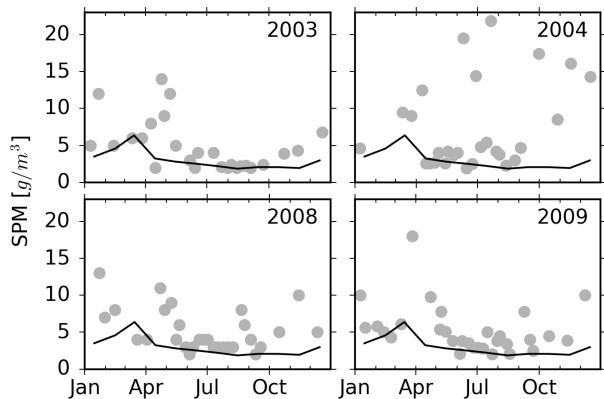

**Figure B2.** SPM concentrations measured (gray dots) and the monthly climatologies used as model forcing (black line) at Noordwijk-10 station (location shown in Fig. 6).

## B2    Phytoplankton Sinking Rates and Vertical Chlorophyll Profiles

Average phytoplankton sinking rates estimated by the model at the surface layers vary considerably across seasons, with higher sinking rates at the offshore areas during summer than in the rest of the year (Fig. B3). Sinking rates at the bottom layer are lower than at the surface, both during spring and summer (Fig. B3). These patterns are driven by the nutrient quota dependence of sinking rates (Eq. A25), and low nutrient quotas at the surface layers in the stratified offshore ares during summer (Fig. 10), caused by depletion of nutrients at the surface layers (Fig. 11). Because of the continuous supply of nutrients from the

sediments, phytoplankton at the bottom layer maintain high nutrient quotas throughout the year, and have therefore lower sinking rates. Range of observed mean sinking rates (-0.92-1.14 m/d) in the Rhine ROFI (Peperzak et al., 2003) is roughly consistent with those estimated by the model. Estimated sinking rates being higher during summer than in spring, and at the surface than at the bottom layer seem to be also roughly consistent with the observations made at the Yangtze River estuary (Guo et al., 2016, Fig.6). However, both in the Yangzte River estuary and in the Rhine ROFI, some observations indicate

higher sinking rates at the deeper layers during spring (Peperzak et al., 2003; Guo et al., 2016). This is explained by the species-specific differences in sinking rates (e.g., sinking rates of Diatoms being higher than Dinoflagellates and *Phaeocystis*), and differences in community composition at the surface (dominated by dinoflagellates or *Phaeocystis*) and bottom layers (dominated by diatoms), therefore, cannot be captured by the presented single species model.

The vertical distribution of chlorophyll qualitatively depends on the formulation of phytoplankton light availability, sinking

and the resource utilization traits of phytoplankton. To demonstrate this, we considered alternative parameterizations regarding

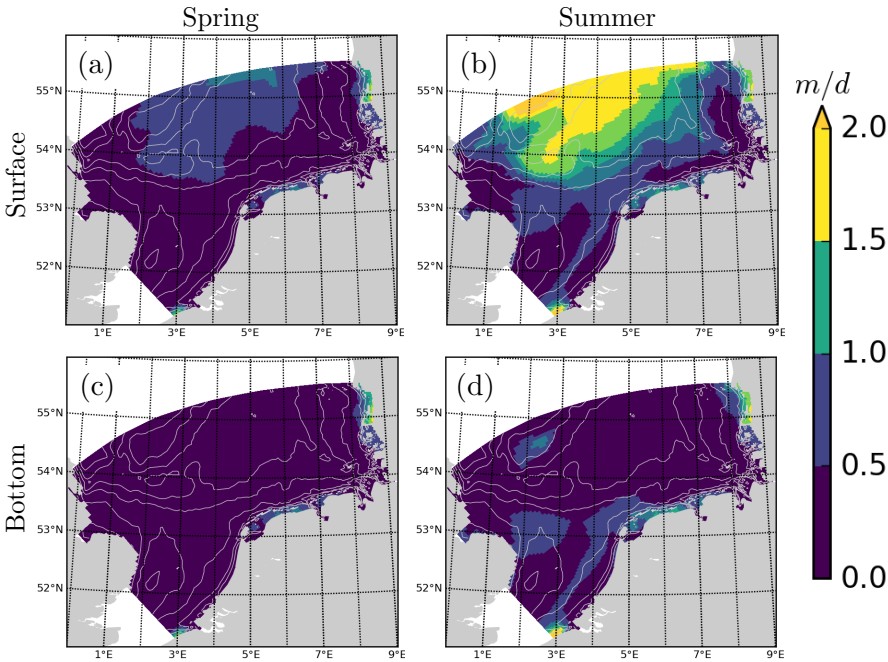

**Figure B3.** Phytoplankton sinking rates (a,b) at the surface layer and (c,d) at the bottom layer, averaged over spring (months 3-5, left) and summer (months 6-8, right) of 2010.

the sinking of phytoplankton and light climate and a non-acclimative model version where the physiology of phytoplankton is fixed (see section B3 for a detailed description), and compare the resulting chlorophyll profiles with the original (reference) model run for 3 example days and at a deep spot inside the German Bight (Fig. B4). The vicinity of this time interval and location (55°N, 5°E) was characterized by an occurrence of a thin chlorophyll layer according to the Scanfish data, as captured
quite realistically by the reference run (Fig. 9).

The model run with constant and low sinking rate also resulted in deep chlorophyll maxima for the first 2 days (Fig. B4a), which is, however not concentrated at the thermocline as in the reference run (Fig. 9), but rather close to the surface with a wider vertical distribution, and on the third day a monotonic profile with the higher values homogeneously distributed within the upper mixed layer. The model with constant and high sinking rate on the other hand, resulted in profiles monotonically
increasing towards the bottom, with overall low concentrations (Fig. B4a). The model run that assumed a spatially constant, low background attenuation values characteristic of clear ocean waters resulted in a sharp increase in chlorophyll concentrations at around 15 meters depth as in the case of the reference run (Fig. B4b), but then the concentrations do not decrease towards the bottom significantly, unlike in the case of the reference run. Higher specific attenuation coefficients resulted in distinct deep chlorophyll maxima in the first 2 days, although about 5 m closer to the surface, and on the 3rd day homogeneous distribution
within the upper mixed layer, again unlike in the case of the reference run (Fig. B4b). Finally with the simplified model with

fixed physiologies for two different parameterizations regarding the allocation to the light harvesting, nutrient acquisition and carboxylation machineries, phytoplankton ends up always being concentrated at the bottom layers (Fig. B4c).

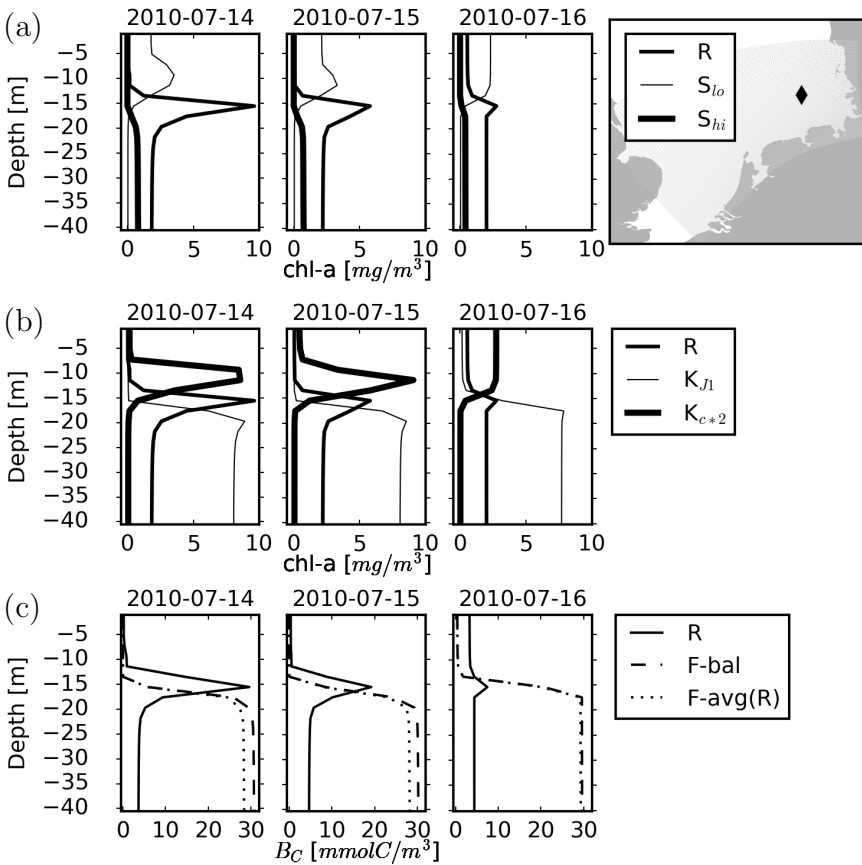

**Figure B4.** Vertical distribution of the simulated phytoplankton chlorophyll(a-b) and carbon (c) for 3 example days in July 2010 at an example spot (55°N, 5°E, shown with the diamond symbol in the inset map) for various model realizations, regarding; **a)** phytoplankton sinking, where R: the reference model (with $w_B^0$=0.2 and $w_B^*$= 3.0 m/d as in Table A3); $S_{lo}$ and $S_{hi}$: phytoplankton sinking rate set to constant ($w_B^*$= 0.0 m/d), and respectively, low ($w_B^0$=0.2) and high ($w_B^0$=3.0) values; **b)** the light climate, where R: the reference model with the light attenuation within the blue-green spectrum (as described by the length scale coefficient $eta_2$ in Eq.2) determined by a background SPM-caused turbidity used as model forcing (see section 2.2.3) and shading by the modeled variables (with specific attenuation coefficients listed in Table A3); $K_{J1}$: low attenuation achieved by setting a small (=23m) background value for $eta_2$ (characterizing clear-water conditions Paulson and Simpson, 1977) and keeping the specific attenuation coefficients as in R; $K_{c*2}$: high attenuation achieved by setting the specific attenuation coefficients twice their original values and keeping the SPM-caused turbidity as in R; **c)** resource utilization traits of phytoplankton, here as represented by R: the reference model with dynamic and optimal allocation of resource utilization traits; and two non-acclimative model versions (explained in Appendix B3 and shown in Fig.14) F-bal: balanced allocations to light harvesting, nutrient acquisition and carboxylation; and F-avg(R): allocation coefficients calculated as the spatio-temporal averages from the reference run.

## B3   Non-acclimative Model

For gaining insight into the relevance of acclimation aspects of the model, we considered a simplified version of the model in which the adaptive and optimality based features of the model were excluded. For transforming the full model to an otherwise equivalent non-acclimative version:

1. Dynamic equations Eq.A7-A8 that describe the allocation of resources to light harvesting ($f_{LH}$), carboxylation ($f_C$) and nutrient acquisition ($1 - f_{LH} - f_C$) traits were excluded.

2. Instead of being optimized, $f_i^A$ was fixed to a constant value of 0.5, implying equal investments into affinity and intracellular transport.

3. Uptake activity function, $a_i$ was replaced with a classical linear function of individual cellular quotas ($a_i = \frac{Q_i - Q_i^0}{Q_i^{max} - Q_i^0}$).

In this simplified, fixed-trait model, $f_{LH}$ and $f_C$, which are dynamic state variables in the full model, become parameters. We considered two conceptual assumptions for assigning their values: 1) balanced allocations to each cellular machinery (referred to as 'F-bal' in Fig. 14 and Fig. B4c), achieved by setting $f_{LH} = f_C = 0.333$ 2) assigning the domain-wide, volume-weighted averages obtained with the acclimative model for a specific time period (referred to as 'F-avg(R)' in Fig. 14 and Fig. B4c), which were, for the year 2010 (for which the results are compared), $f_{LH} = 0.38$ and $f_C = 0.24$ for the year 2010. The further two parameters, namely the upper bounds of nitrogen and phosphorus quotas, were set to $Q_N^{max} = 0.35$ and $Q_P^{max} = 0.04$ such that the resulting range of N:C and P:C ratios are similar to those obtained with the full model for the year 2010, for which the results are compared.

*Author contributions.* OK and KW designed and outlined the study. OK calibrated the model, ran the simulations, performed the majority of analyses and drafted the manuscript. OK and RH created the model setup and prepared model forcing. KW, RH and OK developed the biogeochemical model and wrote the model code. RH and OK prepared the salinity cruise plot. RR planned and carried out the Scanfish measurements in the German Bight. RR, JM and RH handled the Scanfish data and contributed to the preperation of the Scanfish plot. JM developed a SPM climatology for a former model version. All authors participated in revising the manuscript.

*Acknowledgements.* We gratefully acknowledge Markus Schartau (Helmholtz Centre for Ocean Research Kiel) for his contributions to the initial development of MAECS, Carsten Lemmen (HZG) for his help with setting up the supercomputing environment, Sonja van Leeuwen (Centre for Environment, Fisheries and Aquaculture Science) for providing data on riverine fluxes, Sönke Hohn (Leibniz Centre for Tropical Marine Research) and Annika Eisele (HZG) for their assistance in formatting and checking the river data, Fabian Große (University of Hamburg -UH) and Markus Kreus (UH) for providing the boundary conditions for the biogeochemical model and Johannes Pätsch (UH) for providing the SPM climatology. Justus van Beusekom (based on his work in AWI, now HZG), Karen H. Wiltshire (AWI), Annika Grage (NLWKN), Thorkild Petenati (LLUR), Sieglinde Weigelt-Krenz (BSH) are acknowledged for providing monitoring data. Justus van Beusekom (HZG), Fabian Große (UH), Ivan Kuznetsov (HZG), Hermann-Josef Lenhart (UH) and Corinna Schrum (HZG) are acknowledged for their helpful comments during various stages of this work. This is a contribution by the Helmholtz Society through the PACES program. OK, RH and KW were supported by the German Federal Ministry of Education and Research (BMBF) throgh the MOSSCO project. OK and KW were additionally supported by the German Research Foundation (DFG) through the priority program 1704 Dynatrait. The authors gratefully acknowledge the computing time granted by the John von Neumann Institute for Computing (NIC) and provided on the supercomputer JURECA (Jülich Supercomputing Centre, 2016) at Jülich Supercomputing Centre. Comments by three anonymous referees helped significantly improving the paper.

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
