# Peer review of "The acclimative biogeochemical model of the southern North Sea"

_Biogeosciences, 2017_

## Referee Comment (RC1) · Anonymous Referee #1 · 27 Apr 2017

Review of A novel acclimative biogeochemical model and its implementation to the southern North Sea, by Kerimoglu et al.

Summary

The manuscript describes the application of a recently published acclimative phytoplankton growth method in a 3D coupled hydrodynamics-biogeochemical model of the southern North Sea. Model results of a decade are compared with an extensive set of observations, and to some extent discussed in relation to the hydrography of the area.

General comments

This paper represents a significant effort to implement new process descriptions for phytoplankton cell biology in a biogeochemical model, and apply the model to a coastal

sea region. It is reasonably well-written, and, eventually, deserves publication. However, I'm struggling with a substantial number of issues (detailed below) that need to be addressed. Hence, I recommend major revisions, and re-review. Note that given this, I have not assessed the appendix, nor the supplementary material.

Title

I have two problems with the title: 1) as the acclimatisation scheme has been published previously, I would advise against using the word 'novel'; 2) 'implementation' suggests the presentation and discussion of how the acclimatisation method is implemented in the biogeochemical model, which is included in the manuscript, but not related to the application to the SNS. So I would suggest reformulating to, eg., The application of an acclimative biogeochemical model to the southern North Sea.

Structure

The authors should introduce a separate discussion section.

Comparison

In comparing model results with observations, the text is too qualitative, using expressions such as 'compare well', 'reasonable match', and so on, without defining what these are. This should be tightened up and quantified throughout. The same holds for comparison with previous work in the literature: a small subset of earlier biogeochemical modelling work is referenced, and it is suggested that the current model performs better, but without providing the evidence and quantifying the differences. It is also unclear why these studies were selected, and not others.

Logic/interpretation

The logic and interpretation tend to be hand-waving at best, flawed in some cases, and don't always consider multiple options. Examples are listed in the details section below. This needs to be improved. Separating the discussion will help.

Is it really 'better'?

The authors state at several points in the paper that their acclimative phytoplankton growth method is better what's used in more traditional biogeochemical models. However, unfortunately, they fail to provide any proof of this. In the very least, there should be an in-depth, quantified discussion comparing the current results with those of a suitably wide range of 'traditional' models. I get the impression from the manuscript that the 'novel' biogeochemical model was constructed by stripping an existing 'traditional' biogeochemical model of the relevant parts, and replacing these with the acclimative methods. If this is indeed the case, the authors would strengthen the manuscript immensely by providing and discussing a comparison with a similar run with the earlier model version. The authors will also need to discuss the following in a systematic way. More traditional biogeochemical models may lack (to various extents depending on the model) the full suite of acclimatisation as presented here, but they make up for that at least to some extent by representing several types of phytoplankton. This allows for spatial and temporal changes/patterns in phytoplankton composition. One could argue that the new model reflects this with one type with a range of traits, but it presumably has more flexibility in changing these traits over time for the same biomass than could happen in nature (one type of plankton can not change into another). Also, the authors are suggesting that they plan the inclusion of additional phytoplankton types. That would require curtailing the ranges of acclimatisation. Would that throw the baby out with the bath water, or have they already done so and would this be an attempt to get it back in?

Figures

Not all of the figures are clearly readable, and some information is missing.

Fig 3: I suspect that the colour scale is truncated, both at the high and low end, resulting in artificial saturation of the figure. This must be addressed. Also this figure would benefit from using a wider range of colours.

Fig 4. S and T are partly obscured by the dots, the cursive eta and n are barely visible on my printout

Fig 5-7. These are all too small. I can hardly read the axis legends and legends. Names on maps are cluttered.

Fig 8. Does ICES store chlorophyll? If so it would help if this were included.

Fig 9. Re-plot in colour. I can't work out the route taken from the cruise track figure.

Fig 10, 11, 14. The black contours are partially obscured by the dark blue.

Fig 11. I understand that these are surface values. Please also provide the bottom values.

Fig. 12. The colour scale is symmetrical around the centre, making it impossible to distinguish spring and autumn values. Please re-plot.

Grammar and language

Please check the grammar. There are quite a few anomalies that even a grammar checker would pick up (I'm not going to list them all).

Also use past tense to describe the results throughout.

Further detailed comments

p. 1

l. 1. autotroph: autotrophic?

l. 5. is based on novel concepts

l. 11 'sparce measurements'. Not clear what these are.

l. 13. delete prevalently

l. 14 shows significant seasonal and spatial variability

l. 14-16. not clear what is meant here

Section 2.1, title. 'Data' can originate from anywhere, including models. Use 'Observations', apply throughout.

p. 2

l. 18 monitoring stations used here(?)

p. 3

l. 11 in the benthos

l. 14 and rivers considered

l. 19 accessory: access to?

l. 24 give value for flexibility constant. Give rang of i, and values for q_i

l. 29 why use 'B' for phytoplankton (most models use P, and B for bacteria)?

p. 8

l. 6. Other explanations could be that: 1) the river-runoff is too high, or 2) the set of open boundary conditions used for the hydrodynamics and disolved components restricts the amount of flushing, leading to an accumulation of fresh water, nutrients, etc.. Or a combination. Please discuss.

l. 15 Now the rivers do come up, but the sentence is unclear, and I don't understand the link to grid resolution.

l. 18. Trends. These figures are not suitable to identify trends.

l. 19. 'in general well reproduced': this too qualitative, I list it here to present an example, but the paper is littered with these kinds of statements (I will not list them all).

l. 20/21. 'rather realistically represented': another one.

p. 9

Fig 4. There seems to be a 1:1 relationship, but with an anomaly on top. Does the anomaly in T correspond to the low values of S that bend away from the 1:1 line? Does this cluster represent a particular geographic area (front?)? Or a particular event/year (2010?)?

l. 2-3. this should be easy to test?

l. 5. Earlier. Than what?

l. 5. 'mostly well reproduced': difficult to see on the small graphs; quantify.

l. 6. 'probably'. other potential causes?

l. 10. 'is entirely reversed': I don't see this...

l. 11-p10 l. 3: this is discussion

p. 11

l. 1. 'easier': than what?

l. 3. variability matches very well: I don't see this/quantify.

l. 6. 'might be': why?

l. 8. 'typical': give numbers

p. 14

l. 2. grammatically incorrect.

p. 15

l. 6. 'were not able to ... observations': but this model doesn't do this, either...

l. 6-13. Please provide evidence for this.

[Figure]

l. 28. 'intuitively predictable': this is a contradiction in terms.

l. 29-34. This seems a rediculous over-interpretation of a potential contribution by estuarine overturning circulation. There's no evidence of overall higher nutrient concs in bottom waters (fig 8). Providing bottom values in fig 11 will likely support this. What's happening is that the nutrient-rich riverine waters enter/mix with the coastal waters, which are trapped by the coastal density(salinity) front.

p. 16

l. 8-p17 l. 5: It's not very clear what the function and message of this section are.

p. 17

l. 7. higher chlorophyll concentrations

l. 9-12. This is an unfair comparison. The observations in fig 5 are instantaneous, whereas the satellite composites are 3-monthly averaged. It's obvious that the satellite values presented in this way should be lower! This statement requires a proper comparison.

l. 13-14. this sentence trips over the various averages. Reformulate/clarify.

p. 18

l. 8. nutrient and turbidity gradients?

p. 19

l. 11-14. Please provide evidence for this.

p. 20

l. 5. ignorance of: ignoring

Figure captions

fig 2. Fe-P is not in the figure. Explain bAP in the caption.

fig 4. delete 'abbreviated' (2x)

fig 5. Observations (circles) and model estimates (lines) ... correlation coefficients (r),... data points (n)

Fig 10, 11, 14. Specify what the black contour lines represent.

fig 13. Mention that this is a log scale. Explain rho and eta.

---

## Referee Comment (RC2) · Anonymous Referee #2 · 17 May 2017

General comments

This paper addresses the challenging task to combine detailed modelling of intracellular processes in phytoplankton with detailed oceanographic modelling. The paper states that the detailed formulation intracellular processes in phytoplankton leads to improved modelling results for the southern North Sea. It would lead to more variability in vertical patterns and to amplification of the cross-shore chlorophyll gradient, due to the effect of light on chlorophyll to carbon ratios. But these conclusions are not well supported by the results:

- The time series comparison of chlorophyll results with in-situ data suggest that chlorophyll concentrations are systematically over-predicted in spring at many monitoring stations. The validation plots with in-situ data are only presented for other model variables

and not for chlorophyll.

- The validation with satellite data shows also that chlorophyll is systematically over-predicted throughout the model domain during spring. The authors conclude that the satellite data are wrong. This is not supported by any comparison with in-situ data, but the above comparison with time series suggests that the model over-predicts chlorophyll in spring.

- It is unclear what trait effects are included in the model, which are not included in existing models. On page 4, line 20 a few traits are listed (very brief) but in the discussion at page 15 and 17 other effects are mentioned, such as effects on chlorophyll to carbon ratio and sinking rates.

- A critical discussion of the novel aspects of the phytoplankton model is lacking. For example: are the chlorophyll to carbon ratios in spring in a realistic range for spring conditions? How do sinking rates change over the year and how does that relate to observations?

- There is no validation of the light climate (as Kd) included in the manuscript. This would be helpful in explaining differences between the model and observed data. Hence, I recommend major revisions.

Specific comments:

-Page 4, line 20 and equation 1. This part needs to give a complete list of acclimation effects included in the model. It should also describe in words how it works. Like it is written on page 17: " sinking speed of algae in MAECS is inversely related to nutrient quota of cells.". So there are not only effects of nutrients on growth rate (as suggested by eq 1) but also on other aspects. And there are effects of light on chlorophyll to carbon ratio. And does a flexibility constant represent?

- Page 5, caption of Figure 2 mentions Fe-P:P adsorbed in iron-phosphorus complexes. I don't see this in the figure. Or you should refer to bAP in the caption.

- Page 7: Could you please clarify in more detail the source of the ESA-CCI dataset. Is there a website where these data can be downloaded and where we can find validation reports of this dataset?

- Figure 4: the T and S are too small to read and overlap with the dots. - Page 9, line 10. I don't see that the classical seasonal pattern of phosphorus is entirely reversed in the data. This may be partly due to the small size of the figure. But also it may be that phosphorus concentrations in shallow muddy areas of the Wadden Sea are higher during the summer than during winter due to release of phosphorus from anoxic sediments. This does not reverse the seasonal pattern, because there is a classical drop in phosphorus concentrations during the spring bloom.

- Page 9: line 15: "potentially inadequate description of certain processes". Here a more thorough discussion of model functioning is needed. Now the validation data is more critically discussed than the model. I would expect that at location with a measurement frequency of several weeks to months, there is not much smoothing effect in monthly averages. Anyway such effect cannot explain structural differences between model and in-situ data, as shown in Figure 5a: DIN is consistently underpredicted and DIP overpredicted by the model.

- Page 10, Figure 5: the Pearson coefficients in the figures are too small to read. It would be clearer to present them in a table.

- Page 13, Figure 8: Please also include similar figures for chlorophyll. Chlorophyll is the only model variable that is relevant to judge the validity of the novel modeling approach.

- Page 15, lines 5 – 13. The reader has no information to judge whether the sinking speeds in MAECS are more realistic than in other models. I would expect that the variability in the physical model underlying the ecosystem model is the main driver of vertical variability in phytoplankton concentrations. I don't see any information to convince me that "intracellular regulation of nutrient storages and pigmentory material"

plays any role in this.

- Page 17, lines 3-5. This is not an entirely open question. There are some interesting papers about this effect, such as: Burson, Amanda, et al. "Unbalanced reduction of nutrient loads has created an offshore gradient from phosphorus to nitrogen limitation in the North Sea." Limnology and Oceanography (2016) and references therein.

- Page 17, line 10: If you use a data source for validation of the model you cannot conclude that the data are wrong instead of the model. Also the reason that some in-situ measurements in Figure 5 are above 50 is not valid. Figure 5 shows that the majority of the in-situ data is well below 50. So to make a fair comparison between in-situ data and satellite data, you should compare the seasonal averages, also at the offshore stations.

- Page 17: lines 13 – 18. Here you only compare patterns in chlorophyll-c ratios with literature, but not the actual ranges. The numbers in Figure 14 are too small to read so I cannot judge whether the overprediction in chlorophyll in spring (figure 13) is caused by too much phytoplankton biomass or too high chlorophyll to carbon ratios.

- Page 20: Lines 3 – 4. This is an interesting conclusion, but it is not well supported by the results presented in this paper.

Technical comments

The text is too small to read in most figures.

---

## Referee Comment (RC3) · Anonymous Referee #3 · 22 May 2017

The authors present the implementation of a coupled 3D physical-biogeochemical model in the southern North Sea. The model included a detailed description of autotrophic growth explicitly taking into account for photoacclimatation and stoichiometric regulation. Model simulations were validated compared to available data for the period 2000-2010.

The paper is well written and clear and I agree on the importance of correctly understanding and representing physiological mechanisms in biogeochemical models. However, some points need to be clarified and/or added to better support their conclusions.

General comments

1) Model formulation: The authors consider a grazing rate function of prey biomass whatever the phytoplankton species represented. There are potential issues with this

hypothesis as Phaeocystis colonies (that can dominate the spring bloom in some of the coastal stations of the studied area) is not grazed by copepods. This should be modified or/and discussed.

2) Model validation: In general, the model reasonably well reproduced available data. However, it is not clear which criteria is used to determine when observed data are realistically represented or not (e.g. p9 L1). This needs to be clarified.

3) Model exploitation: The mechanistic description of the regulation of phytoplankton composition is pointed as an important process and an improvement compared to other existing models to correctly describe primary producers but also nutrient cycling. However, this is not directly evidenced in the paper based on model results. A comparison of results obtained with and without taking into account for these processes is needed to support this conclusion.

Specific comments:

Figure 4: legend 'T' and 'S' on the dots: not clear

P9 L1: How determine 'realistic' and 'not realistic' results ? (see general comment 2)

P15 L6-8: This is an important result and could be developed and evidenced based on model results (Figure with different parameterization of under-water light climate and sinking rate of phytoplankton for example).

Figure 12: Why N:P variability of model results is always lower than the one observed?

P17 L10-11: This is not so clear for me: Fig 5 also shows an important overestimation of simulated Chl a compared to observation.

P 18 L5: The variability of Chl:C can also partly result from the overestimation of Chl a in the model (see previous comment).

P20 L 3-10: This should be evidenced based on comparison of two simulations (with and without taking account for photoacclimation) (see general comment 3)

---

## Author Comment (AC1) · 23 Jun 2017

We would like to thank all three anonymous referees for their constructive comments and criticism. The referees pointed to the need for a more representative title, improved clarity in text and figures, including a separate discussion section, additional comparison of some model estimates with observations and literature beyond what is already presented, and additional analyses for the justification of some conclusions, in several instances harmoniously. While we agree with most comments as detailed below, we believe some of the suggested extensions require more elaborate analysis than can be included here. To recapitulate, the objectives of the current study are; *i*) gaining insight into the behavior of an acclimative model in a 3D framework for the first time to the best of our knowledge; *ii*) evaluating the skill of the new model system at various spatial and temporal scales, which requires consideration of an extremely diverse array of observation sets. These objectives lead to a wide scope, and generation of a number of research questions that are better treated separately.

We have a remark relevant to all three referees, therefore placed here: upon further examination of our model simulation presented in our original manuscript, we found out that the performance of the hydrodynamical model could be improved by not specifying the momentum fluxes at the open boundaries, and a re-parameterization of the bottom friction. We also realized that, due to a wrong configuration file, the atmospheric nitrogen deposition was not correctly registered during the model initialization in the simulation presented in the original manuscript. A new simulation run for the entire simulation period with the improvements in hydrodynamical model and inclusion of atmospheric nitrogen deposition results in better model performance overall, although not qualitatively affecting our conclusions based on the original manuscript. In a revised version of the manuscript, we thus would like to present the results obtained with this new simulation run.

**Detailed response to Referee #1**

*Title: I have two problems with the title: 1) as the acclimatisation scheme has been published previously, I would advise against using the word 'novel'; 2) 'implementation' suggests the presentation and discussion of how the acclimatisation method is implemented in the biogeochemical model, which is included in the manuscript, but not related to the application to the SNS. So I would suggest reformulating to, eg., The application of an acclimative biogeochemical model to the southern North Sea.*

We would like to thank the referee for this careful observation and thoughtful suggestions. Although the referee is right that the acclimatisation scheme of phytoplankton growth was published previously in a 0D context, the full 3D setup including many model variables is described in this manuscript the first time. However the real 'novelty' of this study is embedding the acclimative phytoplankton growth model in a 3D

coupled physical-biogeochemical framework, considering that similarly complex models have been previously studied in much simpler contexts. We agree that the previous title was not reflecting this, so we will change it as: 'A novel 3D coupled physical-biogeochemical model resolving phytoplankton acclimation and its application to the southern North Sea'.

*Structure: The authors should introduce a separate discussion section.*

We will include a separate discussion section as suggested by the referee.

*Comparison: In comparing model results with observations, the text is too qualitative, using expres- sions such as 'compare well', 'reasonable match', and so on, without defining what these are. This should be tightened up and quantified throughout. The same holds for comparison with previous work in the literature: a small subset of earlier biogeochemical modelling work is referenced, and it is suggested that the current model performs better, but without providing the evidence and quantifying the differences. It is also unclear why these studies were selected, and not others.*

We will use more precise formulations for the evaluation of the model. When referring to literature, we attempted to refer to all recent work on the modelling of a relevant model domain. We will check the literature again for potentially omitted work. A detailed and precise comparison of the performance of our model with other models is not in the scope of our study: as was already expressed in the manuscript, such a comparison requires a dedicated effort with standardized benchmarking data and tools. We will stress the need for regional model data bases that will facilitate such model intercomparisons .

*Logic/interpretation: The logic and interpretation tend to be hand-waving at best, flawed in some cases, and don't always consider multiple options. Examples are listed in the details section below. This needs to be improved. Separating the discussion will help.*

[Figure]

We will try to improve the interpretations along the comments of the referees, and include a separate discussion section.

*Is it really 'better'? The authors state at several points in the paper that their acclimative phytoplankton growth method is better what's used in more traditional biogeochemical models. However, unfortunately, they fail to provide any proof of this. In the very least, there sWe will increase the font sizes where necessary.hould be an in-depth, quantified discussion comparing the current results with those of a suitably wide range of 'traditional' models.*

As explained above, we did not intend to claim that our model performs better than the others, and we had expressed clearly in the manuscript that such would require some dedicated effort and is out of the scope of the current work. We will nevertheless try to formulate more precise expressions for evaluating the model performance.

*I get the impression from the manuscript that the 'novel' biogeochemical model was constructed by stripping an existing 'traditional' biogeochemical model of the relevant parts, and replacing these with the acclimative methods. If this is indeed the case, the authors would strengthen the manuscript immensely by providing and discussing a comparison with a similar run with the earlier model version.*

Referee #3 also suggested a comparison with a non-acclimative model version. We did not start from a traditional model and upgrade to an acclimative one, so we do not have such an earlier model version. However, it is possible to turn-off the acclimative features of the model so we will perform a model run and address this relevant issue in the revised version.

*The authors will also need to discuss the following in a systematic way. More traditional biogeochemical models may lack (to various extents depending on the model) the full suite of acclimatisation as presented here, but they make up for that at least to some extent by representing several types of phytoplankton. This allows for spatial and temporal changes/patterns in phytoplankton composition. One could argue that the new*

*model reflects this with one type with a range of traits, but it presumably has more flexibility in changing these traits over time for the same biomass than could happen in nature (one type of plankton can not change into another).*

The plankton functional type (PFT) models might make up for the unrepresented acclimation processes to some extent, but we are not aware of any study which tested this idea rigorously. We do not really see why our model is presumably more flexible than reality: consider the case of the competition of two species, where the first species, dominant at the beginning, is being gradually replaced by the second until it completely vanishes by the end of the experiment. Throughout this plausible experiment, representation of the traits may completely change, possibly without considerable changes in total biomass. A hypothetically perfect simulation of this experiment by our model in terms of biomass and the average trait representation in the system, might seem to suggest that one plankton type changed into another, which is however just an interpretation and not a limitation inherent to our approach. In conclusion, a comparison of the intracellular Chl:C:N:P ratios observed, e.g., in chemostat experiments, and estimations by a PFT model and our acclimation model might provide valuable insights in this direction, however such a comparison would be beyond the scope of the current study. The relevant discussion of the changes in traits in reality vs. as represented by our model provided in the original manuscript (in P.19, L.13-24) is sufficient in our assessment.

*Also, the authors are suggesting that they plan the inclusion of additional phytoplankton types. That would require curtailing the ranges of acclimatisation. Would that throw the baby out with the bath water, or have they already done so and would this be an attempt to get it back in?*

The plastic response simulated by our model can already be seen at the species level as shown by Wirtz and Kerimoglu (2016). Introduction of further plankton groups for resolving other ecophysiological traits such as silicate limitation and edibility will also allow taxa-specific parameterization of resource utilization traits, which is expected to

further improve the representation spatio-temporal distribution of the overall cellular composition of the phytoplankton.

*Figures: Not all of the figures are clearly readable, and some information is missing. Fig 3: I suspect that the colour scale is truncated, both at the high and low end, resulting in artificial saturation of the figure. This must be addressed. Also this figure would benefit from using a wider range of colours.*

The scale was truncated (from the lower range) on purpose, as doing so helps emphasizing the salinity front. We will mention this in the caption of the figure. We will also use the ('viridis') color scheme used in other contour plots and discrete color levels (as well as in other contour plots), as this enables comparing the location of certain value ranges in the measured and simulated data.

*Fig 4. S and T are partly obscured by the dots, the cursive eta and n are barely visible on my printout*

We will increase the vertical spacing on Taylor diagrams and use larger font size in scatter plots.

*Fig 5-7. These are all too small. I can hardly read the axis legends and legends. Names on maps are cluttered.*

For the plot size, we used the standard width required by Biogeosciences. However we will improve the plots by: 1) using larger fonts, 2) using less intrisuve markers for observations, 3) reworking the maps, such that names do not overlap.

*Fig 8. Does ICES store chlorophyll? If so it would help if this were included.*

Referee #2 raised the same question. Our choice to exclude chlorophyll was based on the fact that the spatio-temporal representativeness of the chlorophyll measurements was much inferior in comparison to that of DIN and DIP data, and consideration that the performance of the model with regard to chlorophyll is separately evaluated based on the satellite and Scanfish measurements (Fig. 9 and Fig. 13).

*Fig 9. Re-plot in colour. I can't work out the route taken from the cruise track figure.*

We will re-plot the Figure in color.

*Fig 10, 11, 14. The black contours are partially obscured by the dark blue.*

We found out that using light-gray contour lines produce better results. We will renew these figures accordingly.

*Fig 11. I understand that these are surface values. Please also provide the bottom values.*

The winter concentrations of DIN and DIP at the bottom are almost identical to the values at the surface and this will be mentioned in the revised manuscript.

*Fig. 12. The colour scale is symmetrical around the centre, making it impossible to distinguish spring and autumn values. Please re-plot.*

Objective of coloring in this figure was mainly to distinguish winter, but it is indeed possible that distinguishing spring and autumn values might be of interest, so we will re-plot with an asymmetric color scheme.

*Grammar and language: Please check the grammar. There are quite a few anomalies that even a grammar checker would pick up (I'm not going to list them all). Also use past tense to describe the results throughout.*

We will go through the text and improve the language.

*Further detailed comments*

We provide more detailed answers to the following selection of comments. We will address each of the other minor issues in the revised version.

*p. 8 - l. 6. Other explanations could be that: 1) the river-runoff is too high, or 2) the set of open boundary conditions used for the hydrodynamics and disolved components restricts the amount of flushing, leading to an accumulation of fresh water, nutrients,*

*etc. Or a combination. Please discuss.*

We thank the referee for this insightful comment. As explained at the beginning of this response letter, we found out not specifying the momentum fluxes led to a better representation of the tidal dynamics, hence, the residual currents and as a consequence, spatial distribution of salinity and other transported variables.

*p. 9 - Fig 4. There seems to be a 1:1 relationship, but with an anomaly on top. Does the anomaly in T correspond to the low values of S that bend away from the 1:1 line? Does this cluster represent a particular geographic area (front?)? Or a particular event/year (2010?)?*

We will include a 1:1 line in the scatter plots. In the new model run, that deviation from the 1:1 relationship is largely resolved, but we will investigate the source of such deviations with respect to certain geographic regions and/or events.

*p. 15 - l. 29-34. This seems a rediculous over-interpretation of a potential contribution by estuarine overturning circulation. There's no evidence of overall higher nutrient concs in bottom waters (fig 8). Providing bottom values in fig 11 will likely support this. What's happening is that the nutrient-rich riverine waters enter/mix with the coastal waters, which are trapped by the coastal density(salinity) front.*

The dynamic effect of horizontal density gradients on residual transport of particulate (organic) matter, which we refer to, is a known feature of shallow, tidal seas, as elaborated in detail with observational and modelling approaches in the cited literature (P.15, l. 31-32). The coastal gradients as a result of accumulation of organic matter, is found along the whole SNS coast but the referee is right, that near the estuaries and the regions of freshwater influence (ROFIs), nutrient concentrations are determined more by the riverine nutrient input. The updated model version, which produces weaker stratification than the previous version, hence weaker differences between the surface-bottom waters overall also suggests a more pronounced importance of the estuaries for the nutrient conditions, therefore we will revise the interpretation of gradients in the

light of the explanation by the referee.

*p. 17 - l. 9-12. This is an unfair comparison. The observations in fig 5 are instantaneous, whereas the satellite composites are 3-monthly averaged. It's obvious that the satellite values presented in this way should be lower! This statement requires a proper comparison.*

Both other referees raised relevant questions. This sentence was actually based on such a point-to-point comparison of the raw satellite (not-averaged) and station data (Fig. R1), which reveals a bias in the form of low concentrations by the satellite data although we recognize that the sentence referred by the referee was not formulated clear enough. In the new version of the simulation, this overestimation problem is largely resolved, therefore such statements will not be necessary in the revised version of the manuscript.

**Detailed response to Referee #2**

*The time series comparison of chlorophyll results with in-situ data suggest that chlorophyll concentrations are systematically over-predicted in spring at many monitoring stations. The validation plots with in-situ data are only presented for other model variables and not for chlorophyll.*

While the chlorophyll concentrations are over-predicted consistently for every year for three out of five stations at the coasts of Netherlands (Fig.6), this is the case in only one out of five stations in the German Bight (Fig.5), while in two others (S. Amrum, Norderelbe) the bias is even negative, so we do not agree that the over-prediction problem was 'systematic'. However, with the updated model run (please see the beginning of the response letter), the simulated chlorophyll concentrations match to the observations better in almost all problematic stations. The Referee #1 also pointed to the need to include the chlorophyll in the validation plots. However, as we stated in our response there, 'our choice to exclude chlorophyll was based on the fact that the

spatio-temporal representativeness of the chlorophyll measurements was much inferior in comparison to that of DIN and DIP data, and because the performance with regard to chlorophyll is separately evaluated based on the satellite and Scanfish measurements (Fig. 9 and Fig. 13)'.

*The validation with satellite data shows also that chlorophyll is systematically over-predicted throughout the model domain during spring. The authors conclude that the satellite data are wrong. This is not supported by any comparison with in-situ data, but the above comparison with time series suggests that the model over-predicts chlorophyll in spring.*

This is an issue mentioned by both other referees. As we responded there, a comparison of the raw satellite data with station data clearly reveals a bias (see Fig.R1). On the other hand, with the new model run the overestimation problem during spring is largely resolved, such a critical discussion of the deviations will not be necessary in the new revision.

*It is unclear what trait effects are included in the model, which are not included in existing models. On page 4, line 20 a few traits are listed (very brief) but in the discussion at page 15 and 17 other effects are mentioned, such as effects on chlorophyll to carbon ratio and sinking rates.*

Such trait effects, along with other process descriptions that are not included in traditional models were described in appendix A1 (e.g., effect of nutrient state of the cells on sinking was described in in Page 21, lines 8-12). We will mention the important processes also in the main text, with necessary links and references.

*A critical discussion of the novel aspects of the phytoplankton model is lacking. For example: are the chlorophyll to carbon ratios in spring in a realistic range for spring conditions? How do sinking rates change over the year and how does that relate to observations?*

The agreement between the coastal pattern displayed by the chlorophyll to carbon ratios estimated by the model and that reported by Alvarez-Fernandez and Riegman (2014) was pointed out in the manuscript (page 17, lines 15-16), but a comparison of the actual values was indeed missing. The Chl:C ratios ranging between 0.01-0.1 gChl/gC at the coastal stations and 0.002-0.02 gChl/gC at the off-shore stations reported by Alvarez-Fernandez and Riegman (2014) envelope our estimated seasonal average values of 0.045 and 0.015 within the respective regions. We will mention this in the text. For the sinking rates of phytoplankton in the southern North Sea as well, we will search the literature for relevant information.

*There is no validation of the light climate (as Kd) included in the manuscript. This would be helpful in explaining differences between the model and observed data.*

The largest source of errors for the representation of the light climate in our study is already known: shading by suspended particulate matter (SPM) was incorporated as a climatological model forcing, which has a rather coarse horizontal resolution (about 20km), and does not represent the vertical heterogeneities as well as inter-annual and sub-daily variations. Therefore, a through evaluation of the light field should encompass various spatio-temporal scales, which is therefore better treated in a separate study, which may be dedicated to improving the representation of the light climate. However, we recognize that the issue is relevant for the spatial distribution of the chlorophyll concentrations and chlorophyll:carbon ratios, which is one of the core findings of this study, therefore we will investigate the possibilities to gain some further insight.

*Specific comments:*

*Page 4, line 20 and equation 1. This part needs to give a complete list of acclimation effects included in the model. It should also describe in words how it works. Like it is written on page 17: " sinking speed of algae in MAECS is inversely related to nutrient quota of cells.". So there are not only effects of nutrients on growth rate (as suggested by eq 1) but also on other aspects. And there are effects of light on chlorophyll to*

*carbon ratio. And does a flexibility constant represent?*

A more detailed description of the model was provided in Appendix A1 (e.g., regarding the effects of nutrient quota of cells on the sinking rates), and a full description of the phytoplankton growth model can be found in Wirtz and Kerimoglu (2016, e.g., regarding the effects of light on chlorophyll to carbon ratio). However, as stated above, we will extend the relevant section in the main text with appropriate references and links.

*Page 5, caption of Figure 2 mentions Fe-P:P adsorbed in iron-phosphorus complexes. I don't see this in the figure. Or you should refer to bAP in the caption.*

Indeed b-AP instead of Fe-P should be referred to in the caption. We will correct this in the revised manuscript.

*Page 7: Could you please clarify in more detail the source of the ESA-CCI dataset. Is there a website where these data can be downloaded and where we can find validation reports of this dataset?*

Information on the source of this data set was provided in the text (Page 4, lines 5-8) but we will include the requested additional information in the revised version.

*Figure 4: the T and S are too small to read and overlap with the dots.*

We will use larger fonts and increase the vertical spacing of the labels and the markers.

*Page 9, line 10. I don't see that the classical seasonal pattern of phosphorus is entirely reversed in the data. This may be partly due to the small size of the figure. But also it may be that phosphorus concentrations in shallow muddy areas of the Wadden Sea are higher during the summer than during winter due to release of phosphorus from anoxic sediments. This does not reverse the seasonal pattern, because there is a classical drop in phosphorus concentrations during the spring bloom*

The observed summer-high instead of the typical winter-high in the S. Amrum station was the basis for the 'reverse seasonal pattern', but we agree that this was overly

vague. We will clarify this.

*Page 9: line 15: "potentially inadequate description of certain processes". Here a more thorough discussion of model functioning is needed. Now the validation data is more critically discussed than the model. I would expect that at location with a measurement frequency of several weeks to months, there is not much smoothing effect in monthly averages. Anyway such effect cannot explain structural differences between model and in-situ data, as shown in Figure 5a: DIN is consistently underpredicted and DIP overpredicted by the model.*

In this sentence, we had already hinted at what might be the relevant processes ('..such as the grazing formulation of zooplankton and the representation of the light climate') but we will extend this in the revised version. Moreover, the smoothing effect in this sentence was about Chlorophyll, and not DIN and DIP. On the other hand, the consistent under-prediction of DIN and over-prediction of DIP was found to be caused by the atmospheric deposition not properly registered during model initialization due to a wrong configuration file -we would like to thank the referee for this careful observation. In the new model version which includes atmospheric deposition, this problem is resolved.

*Page 10, Figure 5: the Pearson coefficients in the figures are too small to read. It would be clearer to present them in a table.*

We will include either a table or a Taylor Diagram for summarizing the model performance against station data in the revised manuscript.

*Page 13, Figure 8: Please also include similar figures for chlorophyll. Chlorophyll is the only model variable that is relevant to judge the validity of the novel modeling approach.*

As explained above, performance of the model for chlorophyll is separately assessed using other data sources, and the representativeness of the chlrophyll measurements available in the ICES data base is much inferior in comparison to that of the DIN and

DIP measurements.

*Page 15, lines 5 – 13. The reader has no information to judge whether the sinking speeds in MAECS are more realistic than in other models. I would expect that the variability in the physical model underlying the ecosystem model is the main driver of vertical variability in phytoplankton concentrations. I don't see any information to convince me that "intracellular regulation of nutrient storages and pigmentory material" plays any role in this.*

We did not intend to claim that the sinking speeds in MAECS are more realistic than other models, and full assessment of such a claim is again beyond the scope of the current study. We argued however, that the formation of thin chlorophyll layers is captured better than some other recent modelling attempts, citing one of those as an example. The reproduction of these structures is sensitive to the dependency of sinking rates on nutrient quotas. Description of this dependency in our model was based on earlier literature as explained in Appendix A1 (page21, lines 8-12, eq. A25). The sentence quoted by the referee is based on the fact that the internal quotas in our model scheme is affected by the allocation of resources to light harvesting (as proxied by pigmentory material) and nutrient acquisition (as proxied by nutrient stores), although we recognize that the sentence was possibly misleading, which we will revise.

*Page 17, lines 3-5. This is not an entirely open question. There are some interesting papers about this effect, such as: Burson, Amanda, et al. "Unbalanced reduction of nutrient loads has created an offshore gradient from phosphorus to nitrogen limitation in the North Sea." Limnology and Oceanography (2016) and references therein.*

We thank the referee for pointing to this study, which we were not aware of. We will extend this section with the findings of Burson et al. (2016) and references therein.

*Page 17, line 10: If you use a data source for validation of the model you cannot conclude that the data are wrong instead of the model. Also the reason that some in-situ measurements in Figure 5 are above 50 is not valid. Figure 5 shows that the*

*majority of the in-situ data is well below 50. So to make a fair comparison between in-situ data and satellite data, you should compare the seasonal averages, also at the offshore stations.*

Although we agree that our sentence here was misleading, the 'bias' implied in this sentence was based on a pointwise comparison of the in-situ data with the satellite estimates (Fig.R1). Nevertheless, such tedious discussion will not be necessary in the revised version, as the overestimation problem is largely resolved with the updated model.

*Page 17: lines 13 – 18. Here you only compare patterns in chlorophyll-c ratios with literature, but not the actual ranges. The numbers in Figure 14 are too small to read so I cannot judge whether the overprediction in chlorophyll in spring (Figure 13) is caused by too much phytoplankton biomass or too high chlorophyll to carbon ratios.*

The over-prediction was caused by high phytoplankton concentrations, which is largely resolved in the new model run while the range of chlorophyll to carbon concentrations remains the same, which is between 0.015-0.045 gChl/gC for the seasonal averages as shown in this figure. We will improve the clarity of Figure 14 and add units.

*Page 20: Lines 3 – 4. This is an interesting conclusion, but it is not well supported by the results presented in this paper.*

We agree with the referee. Based on further elaboration of our model results, we will formulate a more precise statement here.

*Technical comments: The text is too small to read in most figures.*

We will increase the font sizes where necessary.

**Detailed response to Referee #3**

*1) Model formulation: The authors consider a grazing rate function of prey biomass whatever the phytoplankton species represented. There are potential issues with this*

*hypothesis as Phaeocystis colonies (that can dominate the spring bloom in some of the coastal stations of the studied area) is not grazed by copepods. This should be modified or/and discussed.*

Our model indeed does not resolve the differences between phytoplankton taxa with respect to grazing defences, as well as other relevant features such as silicate limitation by diatoms. Extension of the model at this stage to resolve such differences is not feasible. We had mentioned (page 19, line24 - page20, line2) that the model could be extended in the future with multiple phytoplankton groups each of which resolve the acclimation processes as described in this study. We will discuss in more detail the implications of having omitted the taxa-specific features, including the grazing resistance of *Phaeocystis*.

*2) Model validation: In general, the model reasonably well reproduced available data. However, it is not clear which criteria is used to determine when observed data are realistically represented or not (e.g. p9 L1). This needs to be clarified.*

Referee #1 also raised concerns about subjective statements regarding the assessment of the model skill. We will formulate more objective and precise expressions for evaluating the model performance.

*3) Model exploitation: The mechanistic description of the regulation of phytoplankton composition is pointed as an important process and an improvement compared to other existing models to correctly describe primary producers but also nutrient cycling. However, this is not directly evidenced in the paper based on model results. A comparison of results obtained with and without taking into account for these processes is needed to support this conclusion.*

Referee #1 also suggested a comparison with a non-acclimative version. We will address this indeed relevant issue in the revised version.

*Specific comments:*

*Figure 4: legend 'T' and 'S' on the dots: not clear*

We will increase the vertical spacing between the labels and markers and use larger font sizes.

*P9 L1: How determine 'realistic' and 'not realistic' results ? (see general comment 2)*

Again, we will formulate more precise and objective statements for describing model performance.

*P15 L6-8: This is an important result and could be developed and evidenced based on model results (Figure with different parameterization of under-water light climate and sinking rate of phytoplankton for example).*

We will perform suggested sensitivity runs and report the results in the revised version.

*Figure 12: Why N:P variability of model results is always lower than the one observed?*

The variability in the N:P ratios is indeed relevant for a better understanding of this Figure. As mentioned in the text, the high N:P ratio is driven by the riverine fluxes, and actually also high atmospheric N-deposition rates close to the land. With the inclusion of the previously omitted N-deposition rates we indeed obtain a better representation of the high N:P cases within the shallower regions (<10m), although not to a full extent, which might be related with the unrepresented seasonality of atmospheric deposition we use as the model forcing. As also mentioned in the text, the low N:P ratios is consequence of a chain of events: phytoplankton growth, sedimentation, and denitrification and phosphorus release in the benthic layer. The inability of the model to reproduce the low N:P events is therefore presumably due to the coarse representation of the benthic processes by the model. However, with the updated model, we also observed a slight improvement in the reproduction of these low N:P cases, which is probably due to a better representation of the tidal currents in the system with a net effect of a restricted transport of high N:P coastal waters to the off-shore areas. We will extend the discussion of the patterns displayed in Fig. 12 in revised manuscript.

*P17 L10-11: This is not so clear for me: Fig 5 also shows an important overestimation of simulated Chl a compared to observation.*

Both other referees raised similar questions about this sentence. Please see Fig. R1, which indeed suggests a systematic underestimation of the higher range of values by the satellite product, although we agree that the sentence there comparing seasonal average values with snapshot values was not fair, but will nevertheless not be needed in the revised manuscript.

*P 18 L5: The variability of Chl:C can also partly result from the overestimation of Chl a in the model (see previous comment).*

A similar concern was raised by the Referee #2 too. With the updated model results, we obtained lower chlorophyll concentrations overall, although the degree of variability in Chl:C ratios was not affected.

*P20 L 3-10: This should be evidenced based on comparison of two simulations (with and without taking account for photoacclimation) (see general comment 3)*

As stated above, we will address this relevant issue in the revised manuscript.
* * *
[Figure]

NOORDWK70
TERSLG50
Helgoland
NOORDWK10
Sylt
Norderney
TERSLG4
SAmrum
Norderelbe

0.40 + 0.29*x

0.39 + 0.30*x

Y-axis: ESACCI chl

X-axis: station chl

**Fig. 1.** Figure R1: Chlorophyll concentrations measured at the stations vs. estimated by the ESA-CCI images.

---

## Author Response (AR1)

**Response to the Reviews**

We would like to thank all three anonymous referees for their challenging, but constructive comments that helped considerably improving the manuscript. The referees pointed to the need for a more representative title, improved clarity in text and figures, including a separate discussion section, additional comparison of some model estimates with observations and literature beyond what is already presented, and additional analyses for the justification of some conclusions, in several instances harmoniously. We agree with most comments, and followed the suggestions of the referees as documented below, but we believe some of the suggested extensions require more elaborate analysis than can be included here. As now more prominently placed in the introduction, the objectives of the current study are; *i*)exemplifying the integration of a fully dynamic physiological regulation model in a 3D framework for the first time to the best of our knowledge, and (now with the additional analysis), gaining some first insight into the relevance of acclimation *ii*) evaluating the skill of the new model system at various spatial and temporal scales, which requires consideration of an extremely diverse array of observation sets. These objectives lead to a wide scope, and generation of a number of research questions that are better treated separately.

We have a remark relevant to all three referees, therefore placed once again here: upon further examination of our model simulation presented in our original manuscript, we found out that the performance of the hydrodynamical model could be improved by not specifying the momentum fluxes at the open boundaries, and a re-parameterization of the bottom friction. We also realized that, due to a wrong configuration file, the atmospheric nitrogen deposition was not correctly registered during the model initialization in the simulation presented in the original manuscript. A new simulation run for the entire simulation period with the improvements in hydrodynamical model and inclusion of atmospheric nitrogen deposition results in better model performance overall, although not qualitatively affecting our conclusions based on the original manuscript. Therefore, in the revised version of the manuscript, we presented the results obtained with this new simulation run.

**Referee #1**

*Title: I have two problems with the title: 1) as the acclimatisation scheme has been published previously, I would advise against using the word 'novel'; 2) 'implementation' suggests the presentation and discussion of how the acclimatisation method is implemented in the biogeochemical model, which is included in the manuscript, but not related to the application to the SNS. So I would suggest reformulating to, eg., The application of an acclimative biogeochemical model to the southern North Sea.*
We would like to thank the referee for this careful observation and thoughtful suggestions. We changed the title to 'The acclimative biogeochemical model of the southern North Sea'.

*Structure: The authors should introduce a separate discussion section.*
A separate discussion section was included in the revised manuscript.

*Comparison: In comparing model results with observations, the text is too qualitative, using expressions such as 'compare well', 'reasonable match', and so on, without defining what these are. This should be tightened up and quantified throughout. The same holds for comparison with previous work in the literature: a small subset of earlier biogeochemical modelling work is referenced, and it is suggested that the current model performs better, but without providing the evidence and quantifying the differences. It is also unclear why these studies were selected, and not others.*

More precise formulations were used throughout the revised text for the evaluation of the model. Moreover, the correlation coefficients and normalized bias values for the biogeochemical variables are additionally presented in a color-coded table with colors indicating model performance, such that at least the internal consistency can be improved. When referring to literature, we attempted to cover most recent work on the modelling of a relevant model domain. In the revised version, we included a few additional relevant work. A detailed and precise comparison of the performance of our model with other models is not in the scope of our study: such a comparison requires a dedicated effort with standardized benchmarking data and tools as mentioned in the discussion.

*Logic/interpretation: The logic and interpretation tend to be hand-waving at best, flawed in some cases, and don't always consider multiple options. Examples are listed in the details section below. This needs to be improved. Separating the discussion will help.*
In the revised manuscript, the discussion section was separated and in many cases, interpretations were extended and supported by additional material (please see the '**List of All Relevant Changes**' at the end of our response letter).

*Is it really 'better'? The authors state at several points in the paper that their acclimative phytoplankton growth method is better what's used in more traditional biogeochemical models. However, unfortunately, they fail to provide any proof of this. In the very least, there should be an in-depth, quantified discussion comparing the current results with those of a suitably wide range of 'traditional' models.*
As explained above, we did not intend to claim that our model performs better than the others, and mentioned in the manuscript that such would require some dedicated effort and is out of the scope of the current work. As also explained in the relevant discussion, different methodologies and datasets used in different studies make even a qualitative comparison difficult. However, we still claim that the estimates for chlorophyll 'can argued to be at least comparable' to those of the earlier studies. We leave however the final judgment to the reader.

*I get the impression from the manuscript that the 'novel' biogeochemical model was constructed by stripping an existing 'traditional' biogeochemical model of the relevant parts, and replacing these with the acclimative methods. If this is indeed the case, the authors would strengthen the manuscript immensely by providing and discussing a comparison with a similar run with the earlier model version.*
Referee #3 also suggested a comparison with a non-acclimative model version. We did not start from a traditional model and upgrade to an acclimative one, so we do not have such an earlier model version. However, in the revised version, two alternative parameterizations of the non-acclimative version of the model was considered (explained in appendix B3), and the resulting horizontal (Fig.14)  and vertical (Fig.B4) distributions were compared with those of the full model.

*The authors will also need to discuss the following in a systematic way. More traditional biogeochemical models may lack (to various extents depending on the model) the full suite of acclimatisation as presented here, but they make up for that at least to some extent by representing several types of phytoplankton.  This allows for spatial and temporal changes/patterns in phytoplankton composition. One could argue that the new model reflects this with one type with a range of traits, but it presumably has more flexibility in changing these traits over time for the same biomass than could happen in nature (one type of plankton can not change into another).*
The plankton functional type (PFT) models might make up for the unrepresented acclimation processes to some extent, but we are not aware of any study which tested this idea rigorously.  We do not really see why our model is presumably more flexible than reality: consider the case of the competition of two species, where the first species, dominant at the beginning, is being gradually

replaced by the second until it completely vanishes by the end of the experiment. Throughout this plausible experiment, representation of the traits may completely change, possibly without considerable changes in total biomass. A hypothetically perfect simulation of this experiment by our model in terms of biomass and the average trait representation in the system, might seem to suggest that one plankton type changed into another, which is however just an interpretation and not a limitation inherent to our approach. In conclusion, a comparison of the intracellular Chl:C:N:P ratios observed, e.g., in chemostat experiments, and estimations by a PFT model and our acclimation model might provide valuable insights in this direction, however such a comparison would be beyond the scope of the current study. Nevertheless, in the new discussion section, we discuss how the PFT models may represent some physiological acclimation.

*Also, the authors are suggesting that they plan the inclusion of additional phytoplankton types. That would require curtailing the ranges of acclimatisation. Would that throw the baby out with the bath water, or have they already done so and would this be an attempt to get it back in?*
The plastic response simulated by our model can already be seen at the species level as shown by Wirtz and Kerimoglu (2016). Introduction of further plankton groups for resolving other ecophysiological traits such as silicate limitation and edibility will also allow taxa-specific parameterization of resource utilization traits, which is expected to further improve the representation spatio-temporal distribution of the overall cellular composition of the phytoplankton. This is further discussed in the new discussion section.

*Figures: Not all of the figures are clearly readable, and some information is missing.*
*Fig 3: I suspect that the colour scale is truncated, both at the high and low end, resulting in artificial saturation of the figure. This must be addressed. Also this figure would benefit from using a wider range of colours.*
The scale was truncated (from the lower range) on purpose, as doing so helps emphasizing the salinity front. This is now mentioned in the caption of the figure. We also used the color scheme used in also other contour plots and discrete color levels to facilitate comparing the location of certain value ranges in the measured and simulated data.

*Fig 4. S and T are partly obscured by the dots, the cursive eta and n are barely visible on my printout*
In the revised version, the spacing between labels and markers were increased, the text annotation were moved outside the scatter plots, larger font size were used and layout was changed to a single-column format for better visibility and reduced space requirement.

*Fig 5-7. These are all too small. I can hardly read the axis legends and legends. Names on maps are cluttered.*
The plots were improved by 1) using larger fonts, 2) carrying the text annotation outside the panels 3) and reworking the maps, such that names do not overlap.

*Fig 8. Does ICES store chlorophyll? If so it would help if this were included.*
Referee #2 raised the same question. We included chlorophyll in the figure.

*Fig 9. Re-plot in colour. I can't work out the route taken from the cruise track figure.*
We re-plotted the Figure in color. Moreover, we used 'cumulative distance' as the position indicator for a more accurate representation of spatial scales.

*Fig 10, 11, 14. The black contours are partially obscured by the dark blue.*
In the revised version we used light-gray contour lines for all contour plots.

*Fig 11.  I understand that these are surface values.  Please also provide the bottom values.*
The winter concentrations of DIN and DIP at the bottom are almost identical to the values at the surface, which is now indicated in the revised manuscript. In this plot, we now provided the growing season averages due to their relevance.

*Fig.  12.  The colour scale is symmetrical around the centre, making it impossible to distinguish spring and autumn values. Please re-plot.*
This figure was removed from the revised manuscript.

*Grammar and language: Please check the grammar.   There are quite a few anomalies that even a grammar checker would pick up (I'm not going to list them all). Also use past tense to describe the results throughout.*
We attempted to improve the language throughout the text.

*Further detailed comments*
*p. 8 - l.  6.   Other explanations could be that:  1) the river-runoff is too high,  or 2) the set of open boundary conditions used for the hydrodynamics and disolved components restricts the amount of flushing, leading to an accumulation of fresh water, nutrients, etc. Or a combination. Please discuss.*
We thank the referee for this insightful comment. As explained at the beginning of this response letter, we found out not specifying the momentum fluxes led to a better representation of the tidal dynamics, hence, the residual currents and as a consequence, spatial distribution of salinity and other transported variables. Accordingly, such a critical discussion is not necessary anymore.

*p. 9 - Fig 4.  There seems to be a 1:1 relationship, but with an anomaly on top.  Does the anomaly in T correspond to the low values of S that bend away from the 1:1 line? Does this cluster represent a particular geographic area (front?)? Or a particular event/year (2010?)?*
Now a 1:1 line is included in the scatter plots. In the new model run, that deviation from the 1:1 relationship is largely resolved. We found out that a majority of these events are located within the western portion of the model domain, as indicated in the text.

*p. 15 - l.  29-34.   This seems a rediculous over-interpretation of a potential contribution by estuarine overturning circulation.  There's no evidence of overall higher nutrient concs in bottom waters (fig 8). Providing bottom values in fig 11 will likely support this. What's happening is that the nutrient-rich riverine waters enter/mix with the coastal waters, which are trapped by the coastal density(salinity) front.*
The dynamic effect of horizontal density gradients on residual transport of particulate organic matter, which we refer to, is a known feature of shallow, tidal seas, as elaborated in detail with observational and modelling approaches in the cited literature.  However the referee is right, that the estuarine-type circulation does not explain the gradients during winter. We therefore updated our interpretation accordingly. Realizing also that nutrient concentrations during growing season is more important for the discussion of the chlorophyll gradients, we now expanded this figure by including the growing-season nutrient concentrations, and mention the estuarine-type circulation as a passing reference.

*p. 17 - l.  9-12.  This is an unfair comparison.  The observations in fig 5 are instantaneous, whereas the satellite composites are 3-monthly averaged.  It's obvious that the satel-lite values presented in this way should be lower!  This statement requires a proper*

*comparison.*

Both other referees raised relevant questions. This sentence was actually based on such a point-to-point comparison of the raw satellite (not-averaged) and station data, which revealed a bias in the form of low concentrations by the satellite data (Fig. R1 in our earlier response letter). However, upon a contact with one of the leading researchers of the Ocean Color Climate Change Initiative, Dr. Shubha Sathyendranath, we were informed that a newer version (v.3.1) of the dataset was recently published, which is particularly improved with respect to the coastal waters with complex optical properties. Furthermore, we were also informed that bias estimates are available for chlorophyll. Therefore in the revised version of the manuscript, this more suitable and bias-corrected product was used, estimates of which agrees with the station data better than the earlier version, although some deviations still exist (Fig.R1). Moreover, the upper range of chlorophyll concentrations predicted by the updated simulation is now much lower than the older model version. As a result of the improvements in both the observation dataset and simulations, the comparison is much better (Fig.R2 below shows the match for the early and later growing season as were shown in the previous version of the manuscript. Fig.12 in the revised manuscript shows the match for non-growing (months 1-3,10-12) and growing (months 4-9) seasons, for the sake of presenting the full annual coverage, but the skill scores for the early and late growing seasons, as well as for the annual average are shown in the new Table 1. There are however still systematic deviations, which are discussed in the text, now with a more balanced account of source of errors.

[Figure]

**Fig. R1**: Chlorophyll concentrations measured at the stations (x-axis) vs. estimated by the Ocean Color v.3.1 product.

[Figure]

**Fig. R2**: as in Figure 12 in the revised text, but for months 4-6 and 7-9. Comparison of satellite (ESA-CCI, a,b) and MAECS (c,d) estimates of surface chlorophyll concentrations averaged over 2008-2010 and for different seasonal intervals of the year. 2-D histograms (e,f) show the number of occurrence of simulation-satellite data pairs. Gray lines in a-d show the isobaths. Normalized bias ($B^*$), Pearson correlation coefficients ($\rho$), and corresponding number of data points ($n$) are shown on top of scatter plots.

Minor issues:
*p. 1 - l. 1. autotroph: autotrophic?*
A: Done
*p. 1 - l. 5. is based on novel concepts*
A: Replaced with 'based on a set of novel concepts'
*p. 1 - l. 11 'sparce measurements'. Not clear what these are.*
A: Expanded.
*p. 1 - l. 13. delete prevalently*
A: Done
*p. 1 - l. 14 shows significant seasonal and spatial variability*
A: Entire sentence was modified.
*p. 1 - l. 14-16. not clear what is meant here*
A: The sentence was expanded.
*Section 2.1, title. 'Data' can originate from anywhere, including models. Use 'Observations', apply throughout.*
A: Done
*p. 2 - l. 18 monitoring stations used here(?)*

A: Done

*p. 3 - l. 11 in the benthos*

A: Done

*p. 3 - l. 14 and rivers considered*

A: Done

*p. 3 - l. 19 accessory: access to?*

A: Entire sentence was reworded

*p. 3 - l. 24 give value for flexibility constant. Give rang of i, and values for q_i*

A: Done

*p. 3 - l. 29 why use 'B' for phytoplankton (most models use P, and B for bacteria)?*

A: Symbols were chosen to keep the notation consistent with Wirtz and Kerimoglu 2016.

*p. 8 - l. 15 Now the rivers do come up, but the sentence is unclear, and I don't understand the link to grid resolution.*

A: This sentence was removed, as with the updated model version, the discussion became obsolete.

*p. 8 - l. 18. Trends. These figures are not suitable to identify trends.*

A: Here the intention is not to identify trends, but we are merely pointing to the fact that there are no 'obvious' trends, which we believe is important to mention, as generation of trends by the model could have been indicative of problems regarding the balance of source-sink terms and/or flawed fluxes at the open boundaries.

*p. 8 - l. 19. 'in general well reproduced': this too qualitative, I list it here to present an example, but the paper is littered with these kinds of statements (I will not list them all).*

A: removed.

*p. 8 - l. 20/21. 'rather realistically represented': another one.*

A: removed.

p. 9 - l. 2-3. this should be easy to test?

A: Elaboration of various assumptions regarding the composition of organic loads desires a study on its own.

*p. 9 - l. 5. Earlier. Than what?*

A: The sentence continues: 'replenishment of phosphrous relative to nitrogen'.

*p. 9 - l. 5. 'mostly well reproduced': difficult to see on the small graphs; quantify.*

A: These section is re-written.

*p. 9 - l. 6. 'probably'. other potential causes?*

A: This is what we think is the most likely explanation. There are other potential causes, but a discussion of them all is not likely to be beneficial.

*p. 9 - l. 10. 'is entirely reversed': I don't see this…*

A: This discussion was removed from the revised manuscript.

*p. 9 - l. 11-p10 - l. 3: this is discussion*

A: moved to the new discussion section.

*p. 11 - l. 1. 'easier': than what?*

A: sentence was removed

*p. 11 - l. 3. variability matches very well: I don't see this/quantify.*

A: Quantified.

*p. 11 - l. 6. 'might be': why?*

A: Sentence was removed.

*p. 11 - l. 8. 'typical': give numbers*

A: Numbers greatly vary, and depend on the processing (eg., binning, as mentioned in the discussion) so giving specific numbers would be misleading.

p. 14 - l. 2. grammatically incorrect.

A: Corrected

*p. 15 - l. 6. 'were not able to ... observations': but this model doesn't do this, either…*

A: We do not agree with this comment. The referee should note that the emphasis here is the diversity of the deep chlorophyll structures, and the fact that our model can reproduce this diversity.

*p. 15 - l. 6-13. Please provide evidence for this.*

A: A new appendix section (B2) is now included to provide evidence for this paragraph.

*p. 15 - l. 28. 'intuitively predictable': this is a contradiction in terms.*

A: Sentence was removed

*p. 16 - l. 8-p17 l. 5: It's not very clear what the function and message of this section are.*

A: Considering quite a number of additions to the manuscript, we indeed decided to remove this discussion.

*p. 17 - l. 7. higher chlorophyll concentrations*

A: Done

*p. 17 - l. 13-14. this sentence trips over the various averages. Reformulate/clarify.*

A: we reworded the sentence.

*p. 18 - l. 8. nutrient and turbidity gradients?*

A: the turbidity gradients are aligned with, therefore oppose the observed chlorophyll gradient (high at the coastal zone). Therefore the main driver should be high nutrient concentrations.

p. 19 - l. 11-14. Please provide evidence for this.

A: This section was re-written.

*p. 20 - l. 5. ignorance of: ignoring*

A: Done.

*fig 2. Fe-P is not in the figure. Explain bAP in the caption.*

A: Done.

*fig 4. delete 'abbreviated' (2x)*

A: Done.

*fig 5. Observations (circles) and model estimates (lines) ... correlation coefficients (r),... data points (n)*

A: Done.

*Fig 10, 11, 14. Specify what the black contour lines represent.*

A: Done

*fig 13. Mention that this is a log scale. Explain rho and eta.*

A: Done

**Referee #2**

*- The time series comparison of chlorophyll results with in-situ data suggest that chlorophyll concentrations are systematically over-predicted in spring at many monitoring stations. The validation plots with in-situ data are only presented for other model variables and not for chlorophyll.*

The updated model (please see the preamble of the response letter) predicts lower spring blooms in all problematic stations. The Referee #1 also pointed to the need to include the chlorophyll in the validation plots. We included chlorophyll from the ICES dataset in validation plots.

*- The validation with satellite data shows also that chlorophyll is systematically overpredicted throughout the model domain during spring. The authors conclude that the satellite data are wrong. This is not supported by any comparison with in-situ data, but the above comparison with time series suggests that the model over-predicts chlorophyll in spring.*

This is an issue mentioned by both other referees. As we responded to the Referee#1 above, a comparison of the raw satellite data (v.2.0) with station data clearly revealed a bias (Fig.R1 in our previous response letter). The updated (v.3.1) and bias-corrected product agrees better with the station data especially at the upper range (Fig.R1). Moreover, as mentioned above, chlorophyll

estimates of the updated model are also much lower now. Overall, the updated model and updated data set suggest a much better agreement now (Fig.R2 for the early and late growing seasons as previously shown in the manuscript, and Fig.12 for the newly chosen non-growing and the entire growing season in the manuscript). Comparison of the model and satellite estimates do not any longer point to an overestimation problem at the upper range but at the rather lower range, which we discuss in the manuscript.

*- It is unclear what trait effects are included in the model, which are not included in existing models. On page 4, line 20 a few traits are listed (very brief) but in the discussion at page 15 and 17 other effects are mentioned, such as effects on chlorophyll to carbon ratio and sinking rates.*
Model description in section 2.2.1 is extended with explanations of the novel and relevant aspects of the model.

*- A critical discussion of the novel aspects of the phytoplankton model is lacking. For example: are the chlorophyll to carbon ratios in spring in a realistic range for spring conditions? How do sinking rates change over the year and how does that relate to observations?*
The agreement between the coastal pattern displayed by the chlorophyll to carbon ratios estimated by the model and that reported by Alvarez-Fernandez and Riegman (2014) was already pointed out in the manuscript, but we now refer to the values and show that the estimations by the model are in a realistic range. In the new Appendix B2, we show the average sinking rates estimated at the surface and bottom layer of the model (Fig.B3), and compare these values with literature.

*- There is no validation of the light climate (as Kd) included in the manuscript. This would be helpful in explaining differences between the model and observed data.*
The largest source of errors for the representation of the light climate in our study is already known: shading by suspended particulate matter (SPM) was incorporated as a climatological model forcing, which has a rather coarse horizontal resolution (about 20km), and does not represent the vertical heterogeneities as well as inter-annual and sub-daily variations. Therefore, a through evaluation of the light field should encompass various spatio-temporal scales, which is therefore better treated in a separate study, possibly dedicated to improving the representation of the light climate. However, we recognize that the issue is relevant for the spatial distribution of the chlorophyll concentrations and chlorophyll:carbon ratios, which is one of the core findings of this study, therefore we included a new appendix section (B1) where we compared the attenuation of the downwelling irradience estimated by the satellite product and by the model. Moreover, at one station (Noordwijk-10), where the model estimates for chlorophyll are particularly biased and poorly correlated with the observations, we compare the measured SPM concentrations with those from the static forcing dataset and mention these results in the discussion.

*Specific comments:*
*-Page 4, line 20 and equation 1. This part needs to give a complete list of acclimation effects included in the model. It should also describe in words how it works. Like it is written on page 17: " sinking speed of algae in MAECS is inversely related to nutrient quota of cells.". So there are not only effects of nutrients on growth rate (as suggested by eq 1) but also on other aspects. And there are effects of light on chlorophyll to carbon ratio. And does a flexibility constant represent?*
In the revised manuscript, the novel aspects of the model are now explained in the main text. Considering its relevance, description of sinking as a function of nutrient quotas was moved from the appendix to the Section 2.2.1. Please see below for the particular case of sinking speed-nutrient quota relationship.

*- Page 5, caption of Figure 2 mentions Fe-P:P adsorbed in iron-phosphorus complexes. I don't see*

*this in the figure. Or you should refer to bAP in the caption.*
Corrected.

*- Page 7: Could you please clarify in more detail the source of the ESA-CCI dataset. Is there a website where these data can be downloaded and where we can find validation reports of this dataset?*
We have included the requested additional information in the revised version. Validation reports are available on-line from the provided website.

*- Figure 4: the T and S are too small to read and overlap with the dots.*
A: Larger fonts were used, layout was changed for better visibility and spacing between the markers the labels were adjusted.

*- Page 9, line 10. I don't see that the classical seasonal pattern of phosphorus is entirely reversed in the data. This may be partly due to the small size of the figure. But also it may be that phosphorus concentrations in shallow muddy areas of the Wadden Sea are higher during the summer than during winter due to release of phosphorus from anoxic sediments. This does not reverse the seasonal pattern, because there is a classical drop in phosphorus concentrations during the spring bloom*
We removed the discussion of this issue from the revised manuscript, considering the increasing relevance of other issues.

*- Page 9: line 15: "potentially inadequate description of certain processes". Here a more thorough discussion of model functioning is needed. Now the validation data is more critically discussed than the model. I would expect that at location with a measurement frequency of several weeks to months, there is not much smoothing effect in monthly averages. Anyway such effect cannot explain structural differences between model and in-situ data, as shown in Figure 5a: DIN is consistently underpredicted and DIP overpredicted by the model.*
In the revised manuscript, the limitations of the model are discussed more systematically in the new discussion section. However, some of the issues pointed out in the previous version such as this discussion about the success of model at this particular station lost their relevance, partially due to the better model performance, but also considering the amount and weight of the new material included in the manuscript.

*- Page 10, Figure 5: the Pearson coefficients in the figures are too small to read. It would be clearer to present them in a table.*
The text annotations (now including the normalized bias) were moved outside the panels with a larger font size. These numbers are additionally presented in a color-coded table (Table 1), where the colors indicate the skill level.

*- Page 13, Figure 8: Please also include similar figures for chlorophyll. Chlorophyll is the only model variable that is relevant to judge the validity of the novel modeling approach.*
We included chlorophyll in the analysis.

*- Page 15, lines 5 – 13. The reader has no information to judge whether the sinking speeds in MAECS are more realistic than in other models. I would expect that the variability in the physical model underlying the ecosystem model is the main driver of vertical variability in phytoplankton concentrations. I don't see any information to convince me that "intracellular regulation of nutrient storages and pigmentory material" plays any role in this.*
We did not intend to claim that the sinking speeds in MAECS are more realistic than other models,

and full assessment of such a claim is again beyond the scope of the current study. We argued however, that the formation of thin chlorophyll layers is captured better than some other recent modelling attempts, citing one of those as an example. The reproduction of these structures is sensitive to the dependency of sinking rates on nutrient quotas, which is now supported by Fig.B4a. Description of this dependency in our model is based on earlier literature as was previously described in the appendix, but now in Section 2.2.1 of the revised manuscript. As also mentioned there, similar relationships have been employed in previous work. The quoted sentence is based on the fact that the internal quotas in our model scheme is affected by the allocation of resources to light harvesting (as proxied by pigmentory material) and nutrient acquisition (as proxied by nutrient stores), although we recognize that the sentence was possibly misleading, therefore we expanded it (relevant paragraph is now in the discussion section). Moreover, comparisons of the vertical phytoplankton concentrations with a model with quota-independent sinking rates (Fig.B4a), and two fixed-trait (i.e., non-acclimative) parameterizations (Fig.B4c) now provide evidence for this paragraph.

*- Page 17, lines 3-5. This is not an entirely open question. There are some interesting papers about this effect, such as: Burson, Amanda, et al. "Unbalanced reduction of nutrient loads has created an offshore gradient from phosphorus to nitrogen limitation in the North Sea." Limnology and Oceanography (2016) and references therein.*
Realizing that spatio-temporal variations in the N:P ratios requires more detailed elaboration, we decided to remove the figure (previously Fig.12) that was relevant to this comment. However, we still refer to this interesting phenomena, and refer to the relevant paper pointed out by the referee.

*- Page 17, line 10: If you use a data source for validation of the model you cannot conclude that the data are wrong instead of the model. Also the reason that some in-situ measurements in Figure 5 are above 50 is not valid. Figure 5 shows that the majority of the in-situ data is well below 50. So to make a fair comparison between in-situ data and satellite data, you should compare the seasonal averages, also at the offshore stations.*
Although we agree that our sentence here was misleading, the 'bias' implied in this sentence was based on a pointwise comparison of the in-situ data with the satellite estimates (Fig.R1 in the previous response letter). Nevertheless, such tedious discussion will not be necessary in the revised version. With the newer satellite product more suitable for the coastal waters (Fig. R1 above)and the updated model, the comparison now is much metter (see also Fig. R2 above for the periods considered in the previous version)

*- Page 17: lines 13 – 18. Here you only compare patterns in chlorophyll-c ratios with literature, but not the actual ranges. The numbers in Figure 14 are too small to read so I cannot judge whether the overprediction in chlorophyll in spring (figure 13) is caused by too much phytoplankton biomass or too high chlorophyll to carbon ratios.*
The over-prediction was caused by high phytoplankton concentrations, which is largely resolved in the new model run while the range of chlorophyll to carbon concentrations remains the same, which is between 0.015-0.045 gChl/gC for the seasonal averages as shown in this figure. We improved the clarity of this figure (now Fig.13, previously Fig.14) by using larger font sizes and discrete color levels and adding the previously omitted units.

*- Page 20: Lines 3 – 4. This is an interesting conclusion, but it is not well supported by the results presented in this paper.*
In the revised version, the new Fig.14 now provides direct evidence for the stronger gradients in phytoplankton concentrations in the acclimative model in comparison to the fixed-trait parameterizations (described in appendix B3), and how these gradients are further strengthened by

the higher chlorophyll:carbon ratios at the turbid, coastal sites.

*Technical comments*
*The text is too small to read in most figures.*
In the renewed figures larger font sizes were used, and layouts were changed for better visibility.

**Referee #3**

*1) Model formulation: The authors consider a grazing rate function of prey biomass whatever the phytoplankton species represented. There are potential issues with this hypothesis as Phaeocystis colonies (that can dominate the spring bloom in some of the coastal stations of the studied area) is not grazed by copepods. This should be modified or/and discussed.*
Our model indeed does not resolve the differences between phytoplankton taxa. Extension of the model at this stage to resolve such differences is not feasible. In the new discussion section, we extensively discuss the limitations of the model, including the necessity of resolving the ecological traits of various plankton groups, such as the grazing resistance of *Phaeocystis*.

*2) Model validation: In general, the model reasonably well reproduced available data. However, it is not clear which criteria is used to determine when observed data are realistically represented or not (e.g. p9 L1). This needs to be clarified.*
Referee #1 also raised concerns about subjective statements regarding the assessment of the model skill. Now the performance metrics are explicitly referred to, correlation coefficient and relative bias (now provided also for the station comparisons). Moreover, the new Table 1, listing the skill scores for DIN, DIP and chlorophyll for all stations and ices data is color-coded, where the colors indicate model skill.

*3) Model exploitation: The mechanistic description of the regulation of phytoplankton composition is pointed as an important process and an improvement compared to other existing models to correctly describe primary producers but also nutrient cycling. However, this is not directly evidenced in the paper based on model results. A comparison of results obtained with and without taking into account for these processes is needed to support this conclusion.*
Referee #1 also suggested a comparison with a non-acclimative version. In the revised manuscript, we considered a model with fixed-trait parameterizations (appendix B3) and compared the horizontal (Fig.14) and vertical distributions (Fig.B4c) resulting from this model with those from the fully acclimative one.

*Specific comments:*
*Figure 4: legend 'T' and 'S' on the dots: not clear*
The spacing between labels and markers were adjusted and larger font sizes were used

*P9 L1: How determine 'realistic' and 'not realistic' results ? (see general comment 2)*
As explained above, by often referring explicitly to the skill scores, and providing the color-coded table we at least hope to have improved the internal consistency for evaluating the model performance.

*P15 L6-8: This is an important result and could be developed and evidenced based on model results (Figure with different parameterization of under-water light climate and sinking rate of phytoplankton for example).*
The suggested sensitivity runs were performed and reported (appendix B2,Fig.B4)

*Figure 12: Why N:P variability of model results is always lower than the one observed?*
Considering the increasing content of the manuscript, and realizing that the variability in N:P requires a more detailed elaboration, we decided to leave this topic out.

*P17 L10-11: This is not so clear for me: Fig 5 also shows an important overestimation of simulated Chl a compared to observation.*
Both other referees raised similar questions about this issue. As can be seen in Fig.R2 above, the agreement between the satellite and model estimates is much improved. The improvement was a result of 1) lower chlorophyll concentrations estimated by the updated model; 2) using a newer version of the satellite product (v3.1 instead of v2.0), which is more suitable for coastal waters with complex optical properties, and indeed shows a better agreement with the station data (Fig. R1). Moreover, at many stations in Fig.5, chlorophyll estimates are also lower by the updated model.

*P 18 L5: The variability of Chl:C can also partly result from the overestimation of Chl a in the model (see previous comment).*
A similar concern was raised by the Referee #2 too. With the updated model results, we obtained lower chlorophyll concentrations overall, although the degree of variability in Chl:C ratios was not affected.

*P20 L 3-10: This should be evidenced based on comparison of two simulations (with and without taking account for photoacclimation) (see general comment 3)*
In the revised version of the manuscript, we considered a fixed-trait (non-acclimative) model (appendix B3), and compare the horizontal and vertical distributions.

**List of All Relevant Changes**

1. With the approval of Editor, Rolf Riethmüller was included as a co-author.
2. Simulation results were repeated with an updated model:
    i. more realistic bottom roughness parameterization (constant, $z_0$=1mm) was employed.
    ii. at the open ocean boundaries, only the surface heights were nudged, while the momentum fluxes were relaxed.
    iii. configuration for atmospheric deposition fluxes were fixed.
3. A figure was removed (formerly Fig.12).
4. A new figure (Fig.14) was introduced.
5. A new table is included for summarizing model skill scores.
6. A new Discussion section was included.
7. A new appendix section (B), with 3 subsections were included:
    i. (B1): validation of light climate, (includes 2 new figures, B1,2).
    ii. (B2): phytoplankton sinking, (includes 2 new figures, B3,4) .
    iii. (B3): description of the non-acclimative model version .
8. A newer version of the ESA CCI-oceancolor data was used in Fig. 12 (which formerly used to be Fig.13).
9. In Fig.9, 'cumulative distance' is used instead of the 'cast-id' in the previous version and the figure was colored.
10. Layout of Fig.4, Fig.8, Fig.10 were changed for better visibility and overall consistency.
11. In Figures 3,4,8,9 color scheme was changed for improved overall consistency.
12. 2 new panels were added both to Fig.10 (March and October values) and Fig.11 (averages for months 4-9).

[revised manuscript text omitted]

---

## Author Response (AR2)

Dear Prof. Middelburg,

thank you once again for handling our manuscript, and your helpful corrections. Below is a detailed response to all your corrections, and a few other minor corrections I identified in the meanwhile.

Best regards,
Onur Kerimoglu

**Response to the Editor's Review:**
-p.3, l. 12: replace on the other hand with however (there is no on the other hand).
Done.
-p.3, l. 22: For a 11-year hindcast...
Done.
-p.3, l. 30: measurements
Done.
-p. 4: figure caption: ... location of stations/rivers...
We show only rivers in this figure. Stations are shown in Fig.5-7.
-p. 4, l. 8, bias-corrected
Done.
-p.4, l. 11: lower trophic food-web dynamics,..
done.
-p.6, l. 20: ..is prescribed...
Done.
-p.10, l.11: mean bias at aroung (i.e. delete and correlation coefficients).
The sentence is reformulated as 'normalized mean bias ($\leq 12\%$) and correlation coefficients at around 0.6–0.7'
-p.27, l. 12. ...A4). The diffusive flux of DIM....
Done
-Table A5: P-sorption
Done.
-p. 33, ...dampened relative to that ...
Done.
-p. 34: the first line of text mentions 2008-2009, while figures shows four years.
'2008 and 2009' changed to: '2003, 2004, 2008 and 2009'
-p. 34, l. 16: higher than those of Dinoflag....
Done.
-p. 37, l 12: 0.333, 2
Done.

**Additional Corrections:**
-P8,L22: 'NOx and NH3' changed to 'oxidized and reduced nitrogen' (pointed out by Daniel Neumann, now included in the acknowledgments)
-Corrected the bibliographic information of Wirtz&Kerimoglu 2016
-Added the doi for several references

[revised manuscript text omitted]